# DATA-JUICER SANDBOX: A COMPREHENSIVE SUITE FOR MULTIMODAL DATA-MODEL CO-DEVELOPMENT

## ABSTRACT

The emergence of large-scale multimodal generative models has drastically advanced artificial intelligence, introducing unprecedented levels of performance and functionality. However, optimizing these models remains challenging due to historically isolated paths of model-centric and data-centric developments, leading to suboptimal outcomes and inefficient resource utilization. In response, we present a novel sandbox suite tailored for integrated data-model co-development. This sandbox provides a comprehensive experimental platform, enabling rapid iteration and insight-driven refinement of both data and models. Our proposed "Probe-Analyze-Refine" workflow, validated through applications on state-of-the-art LLaVA-like and DiT-based models for image-to-text and text-to-video tasks, yields significant performance boosts, such as topping the VBench leaderboard. We also uncover fruitful insights gleaned from exhaustive benchmarks, shedding light on the critical interplay between data quality, diversity, and model behavior. All codes, datasets and models are open-sourced to foster future progress.

## 1 INTRODUCTION

The advent of multimodal generative models has revolutionized artificial intelligence, pushing the boundaries of functionality and creativity across various domains (OpenAI, 2024a;b; Wang et al., 2024b). Recognizing the pivotal role of training data in shaping model performance, there are fast-growing efforts to curate datasets of larger scales and higher quality (Jakubik et al., 2024).

However, the development trajectories of these models and datasets have historically diverged, guided more by intuition than by systematic co-development methodologies. Recent advances in enhancing multimodal generative models tend to be either model-centric or data-centric, rarely bridging the two aspects cohesively. For example, model-centric methods focus on algorithmic enhancements and architectural innovations under fixed data priors, while data-centric strategies usually concentrate on processing and cleaning datasets independently of specific model training contexts (Qin et al., 2024). Both approaches usually suffer from a lack of systematic guidance and cooperative synergy, relying heavily on heuristic exploration and single-perspective expertise. This fragmented landscape presents a significant barrier to achieving optimal model performance, as the interplay between data characteristics and model capabilities remains largely underexploited.

Moreover, the practical implementation of multimodal generative models is further complicated by infrastructure constraints, escalating computational costs, and the accelerating pace of development cycles (Xu et al., 2024c). In the age of large-scale models with rapidly growing model parameters and dataset sizes, the processes of data processing and model training become increasingly resource-intensive, demanding substantial time and computations. Due to the absence of cost-effective platforms that simplify and speed up data-model co-development, researchers and developers often face the dilemma of prioritizing result-driven development at the expense of thorough, insight-led exploration. This deficiency hinders the iterative refinement for both domains, leading to sub-optimal outcomes as improvements in one domain are hard to inform, apply and enhance each other directly.

To fill this gap, we introduce the Data-Juicer Sandbox, a comprehensive suite for facilitating the co-development of multimodal data and generative models. Building upon an open-source data processing system tailored for multimodal generative models, Data-Juicer (Chen et al., 2024a), our sandbox suite further integrates a wealth of off-the-shelf components optimized for usability and compatibility with existing model-centric infrastructures. Collectively, it offers flexibly customizable

Figure 1: Overview of the Data-Juicer Sandbox Laboratory. The workflow involves four stages, each allowing flexible customization at different levels within the data-model co-development lifecycle.

orchestration from different levels including end-to-end workflows, specific development behaviors, and underlying data-model development capabilities. Within the sandbox laboratory, users are empowered to rapidly explore different data pipelines and model configurations on cost-controlled datasets and models. This accelerates the discovery of insightful patterns and informed decision-making, ultimately paving the way for scalable, resource-efficient data-model development.

To exemplify the efficacy of our sandbox, we propose a "Probe-Analyze-Refine" workflow, meticulously crafted to explore the synergies between data processing operators (OPs), target model metrics, and the scalability of these enhancements. We apply this workflow to three cutting-edge models: Mini-Gemini (Li et al., 2024b), an LLaVA-inspired model for image-to-text generation, EasyAnimate (Xu et al., 2024b) and T2V-Turbo (Li et al., 2024a), two Diffusion Transformer based models for text-to-video generation, and a CLIP model Gadre et al. (2023) for image-text foundation model training. Thanks to the sandbox's capabilities, we attain significant advancements in data quality and model performance, such as achieving top spot on the VBench (Huang et al., 2024) leaderboard, outperforming strong competitors such as Gen-3 (RunwayML, 2024) and VEnhancer (He et al., 2024a). These achievements are underpinned by a series of insights linking more than 40 data processing OPs and 30 model metrics, including analysis of the fine-grained impact of data processing for model training, the delicate balance between data diversity and model performance, and the strategic optimization of data scaling for enhanced model-data co-development.

Our contributions can be summarized as follows:

- Innovative Sandbox Suite: To the best of our knowledge, this is the first open-source sandbox suite tailored for co-development between multimodal data and generative models, rendering experimental exploration in this field more insightful, systematic, convenient and reusable.
- Effect-Proven Workflow: We present a new progressive workflow for data-model co-development and substantiate its impact through extensive empirical evidence such as achieving new top-tier performance in image understanding and video generation tasks.
- Practical Guidance: We conduct extensive experiments on benchmarking the effects of dozens of data processing operators and model metrics, providing fruitful and valuable insights toward further advancements in multimodal generative models.

## 2 RELATED WORKS

**Model-Centric Progress in Multimodal Large Models.** Multimodal large models have gained prominence for their remarkable capabilities (OpenAI, 2024a;b). Advances in training algorithms (Caffagni et al., 2024; Li et al., 2024a) and model architectures (He et al., 2024a; Yin et al., 2024) have fueled this interest. Transformer scaling remains a prevalent approach (Xu et al., 2023), though high computational demands and optimization challenges often restrict insights to specific datasets and create a gap in understanding how implicit data biases affect model performance.

**Trends in Data-Centric Development.** Recently, a shift towards data-centric development has emerged (Jakubik et al., 2024), emphasizing data handling as key to efficacy of large models such as CLIP (Gadre et al., 2023). Despite the increasing recognition of data processing, the heterogeneous nature of multimodal data leads to predominantly heuristic approaches (Long et al., 2024), underscoring the pressing need for more systematic methodologies for data-model co-development.

**Open-Source Infrastructure.** The ecosystem for multimodal model development has expanded with fruitful open-sourced frameworks (Wolf et al., 2020; Liu et al., 2023). However, contributions to multimodal data infrastructure often consist of raw datasets and preprocessing scripts, lacking standardized practices. Most existing data processing frameworks are tailored for single-modal data (TogetherAI, 2023), highlighting the early stage of multimodal data development. To address these gaps, our work presents an innovative intermediary layer that connects advanced model infrastructures with the Data-Juicer system, facilitating better co-development between models and data.

More detailed discussions on related works can be found in Appendix A.

# 3 THE PROPOSED DATA-JUICER SANDBOX LABORATORY

## 3.1 MOTIVATION AND DESIGN

**Why do we need data-model co-development?** In the era of large models, the development of both data and models necessitates collaboration involving numerous algorithm researchers and system engineers. Training data for large models is often highly heterogeneous in terms of quality, context, type, and timeliness. The processing and mixing of these datasets, known as *data recipes*, are complex and varied (Ge et al., 2024). The scale of data amplifies the stakes for refinement attempts on both data and models, imposing substantial computational and time burdens. Traditional data-centric or model-centric strategies fall short by optimizing in isolation, leading to *diminished overall efficiency* and *resource misallocation* (Qin et al., 2024). When either component requires adjustment, the dual optimization challenge inflates costs, as one part may have already reached near-optimal status.

**Why do we need a sandbox laboratory?** Given the high cost associated with iterative development, existing methods often resort to heuristic approaches for improving data or models. For example, scaling up "cleaned" datasets can be problematic because determining what constitutes a "clean" dataset and measuring its quality qualitatively remains a challenge. Further, the impact of iterations on data and models is difficult to attribute due to numerous influencing factors and considerable engineering effort required. More insightful solutions are thus needed for data-model co-development.

A unified sandbox environment offers controlled experimentation with low overhead, high transferability, and guided optimization. It permits users to swiftly iterate and optimize data recipes using cost-controlled datasets and models, with insights readily scalable to full-scale production environments, thus addressing larger computational demands of data processing and model training.

**The three-layer orchestration of Data-Juicer sandbox.** To accommodate the multifaceted workflows intrinsic to data-model co-development, we design a versatile, user-friendly sandbox laboratory, as depicted in Fig. 1. The laboratory incorporates a spectrum of components for activities such as data analysis, filtering, recipe optimization, model training, inference, and evaluation, all orchestratable via configuration files. The architecture is stratified into three tiers: bespoke end-to-end *workflows*, generic development *behaviors*, and foundational data-model development *capabilities*.

The top tier delineates co-development workflows executed sequentially across four phases: probing data/models, refining data recipes, executing data/model operations, and evaluation. The sequence of tasks within each phase is adjustable through an input configuration file, permitting users to leverage pre-established and effect-proven workflows or customize their own with ease. Moreover, users can flexibly introduce or innovate classes at the capabilities level (such as novel models, metrics, or data processing algorithms) and behavior hooks (like multidimensional data quality assessments and adaptive adjustments based on multiple probe outcomes) interchangeably.

This design allows for streamlined configuration and reuse of established infrastructure. Importantly, it expedites the prototyping of data and model development solutions, integrating actionable and measurable capabilities for swift feedback derivation and informed decision-making.

## 3.2 FLEXIBLE CAPABILITY FACTORY AND BEHAVIOR HOOKS

**Advanced data processing.** Within the sandbox, we design numerous factory classes and behavior hooks for data processing and evaluation, simplifying and unifying interfaces provided by the open-source system Data-Juicer. Users gain great flexibility to leverage over 100 feature-rich OPs and dedicated tools for efficient data analysis, evaluation, filtering, modifying and synthesis.

From an *actionable* perspective, classes for data processing invoke off-the-shelf OPs from Data-Juicer, such as Filters and Mappers, enabling accelerated and scalable data processing through system

optimization and parallel support. Besides, classes for refining data recipes automatically utilize dedicated tools from Data-Juicer, such as adjusting percentile distributions to differentiate data subsets or applying k-sigma rules to filter out outlier data points. From a *measurable* perspective, classes for data analysis and evaluation are also provided. They encapsulate various observational capabilities within Data-Juicer, enabling efficient computations of target metrics such as text perplexity, video aesthetic value, and image quality scores based on GPT-4V API calls, providing common statistical values, including mean, variance, and percentiles.

**Streamlined model advancement.** Similarly, model development classes integrate state-of-the-art open-source tools and libraries, streamlining interfaces and enhancing usability for rapid and user-friendly development experiences. Classes for actionable capabilities encompass diverse training functionalities, such as Mini-Gemini for image-to-text, EasyAnimate and T2V-Turbo for text-to-video, and ModelScope for general generative models. Classes for measurable capabilities offer evaluation interfaces like VBench and FVD (Unterthiner et al., 2019) for synthesized video assessment, GPT-4 (OpenAI, 2023) for generated text ranking, and TextVQA (Singh et al., 2019), MMBench (Liu et al., 2023), and MME (Fu et al., 2023) for image-to-text synthesis evaluation.

Model development is facilitated through the use of subprocesses to call bash scripts of integrated libraries, with code and parameter adjustments ensuring a cost-controlled environment where, for example, a single GPU card can complete a training session within one day, providing rapid feedback.

## 3.3 A Probe-Analyze-Refine Workflow for Data-Model Co-Development

To demonstrate the usage of the sandbox laboratory, here we elucidate a built-in systematic workflow illustrated in Fig. 2. We commence with a series of cost-effective contrastive explorations answering several key questions that can enhance large-scale data-model co-development:

(1) Considering the variability in data sources, models, and downstream tasks, we initially seek to ascertain *which data processing operators (OPs) contribute most effectively to enhancing specific target model metrics.* This involves creating equal-size data pools, each processed uniquely by a singular OP and subsequently ranked by data metrics. Models are trained on these data pools, enabling us to perform an in-depth analysis of OP effectiveness and its correlation with model performance across various quantitative and qualitative indicators.

(2) Guided by insights derived from the most impactful OPs that ranked highest by model metrics, we proceed to study *whether these OPs can be effectively combined into data recipes and scaled up.* To facilitate this exploration, we establish a hierarchical data pyramid, wherein data pools are categorized across different tiers based on their ranked model metric scores. This stratification also elucidates the viability of OP combinations when scaled with increased data volumes.

(3) Finally, we delve into optimizing data utilization through a dual analysis focusing on *duplication* and *diversity.* We will assess whether the training process would benefit more from the repeated use of high-quality data pools or from the inclusion of lower-quality data to expand the overall data pool. Through a systematic examination of cost-controlled experiments, we can formulate optimized data recipes and datasets, which are then leveraged in more resource-intensive training sessions to cultivate models with superior performance.

### 3.3.1 Single-Operator Data Pools

Starting with an initial dataset $\mathcal{D}$, we define a single-OP data pool $\mathcal{P}_i$ as the dataset processed exclusively by the $i$-th OP ($\mathcal{OP}_i$) available in Data-Juicer as $\mathcal{P}_i = \mathcal{DJ}[\mathcal{OP}_i(\rho_i)](\mathcal{D})$, where $\mathcal{DJ}$ denotes the data-processing function implemented by Data-Juicer, and $\rho_i$ is the hyper-parameters governing the operation of $\mathcal{OP}_i$. The OPs of Data-Juicer can compute specific statistics for every data pool, and apply threshold criteria to selectively filter or modify the data based on these statistics. Within this workflow (demonstrated in the upper left part of Fig.2), the initial dataset $\mathcal{D}$ is processed by $N$ studied filter OPs, and each $\mathcal{P}_i$ is sorted by statistical measures and segmented into three equal-sized data pools $\mathcal{P}_{i,\text{low}}$, $\mathcal{P}_{i,\text{middle}}$ and $\mathcal{P}_{i,\text{high}}$, representing data with low, middle and high stats, respectively. Besides, $\mathcal{D}$ is randomly sampled to serve as a control group $\mathcal{D}_{\text{random}}$ such that all the $3N + 1$ data pools have the same data size. This stratification fosters discriminative insights across varying degrees of data processing intensity.

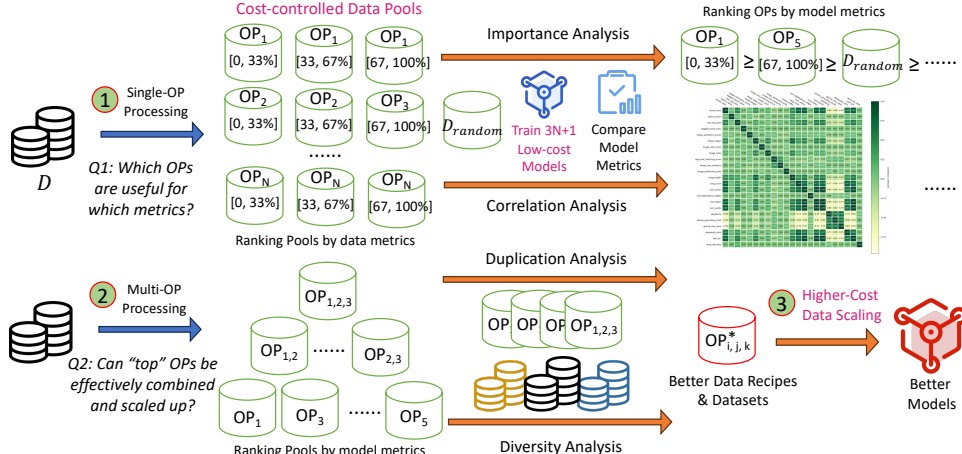

Figure 2: A probe-analyze-refine workflow for systematic data-model co-development.

Subsequently, models are trained independently on each data pool, undergoing comprehensive evaluation across multiple performance metrics. Throughout the training, we uphold consistent hyper-parameters and ensure $\mathcal{D}_{\text{random}}$ is of substantial size to yield a reliable and robust average performance across all downstream tasks. This setup enables the identification of best-performing OPs that universally excel or excel under specific evaluation criteria.

Importantly, we adhere to a cost-conscious design and implement strategies to minimize expenses, such as adopting efficiency-enhancing techniques like LoRA and limiting training iterations to only those necessary for achieving reasonable performance improvements (detailed in Appendix B.6). It enables one trial of the entire experimental workflow to be completed affordably within a single day.

### 3.3.2 MULTI-OPERATORS DATA POOLS

Having explored the individual impact of each OP, our next step logically involves understanding the dynamics when multiple OPs are applied sequentially, as seen in a data "recipe" (illustrated in the bottom left part of Fig.2). This sequence of OPs is referred to as $\mathcal{S}$. Our aim is to discern whether these OPs complement or counteract each other's effects. To achieve this, we extend the data pool construction methodology from previous individual OP scenarios to multi-OP recipe scenarios: $\mathcal{P}_{\mathcal{S}} = \big(\mathcal{DJ}[\mathcal{OP}_i(\rho_i)] \circ \mathcal{DJ}[\mathcal{OP}_j(\rho_j)] \circ \cdots \circ \mathcal{DJ}[\mathcal{OP}_k(\rho_k)]\big)(\mathcal{D})$, where $i, j, ..., k \in \mathcal{S}$.

The number of possible multi-OP data pools grows exponentially with the addition of each new OP, necessitating a strategic selection of combinations to explore. Drawing on insights from single-OP experiments, we propose two pragmatic strategies: (1) combining "Top" OPs based on their progressively diminishing impacts on model performance and (2) clustering OPs based on their Pearson correlation coefficients and then combining "Top" OPs within each category. More details and empirical results on these strategies can be found in Appendix B.5 and C.3 respectively.

In alignment with our methodology for single-OP data pools, we consistently train and evaluate models for each selected multi-OP data pool, while ensuring the process remains cost-effective by maintaining a small pool size still yielding reasonable and distinguishable training outcomes.

### 3.3.3 PYRAMID-SHAPED DATA POOLS

Incorporating a greater number of OPs in a recipe may lead to enhanced data quality; however, the resultant data pool volume decreases exponentially with each additional OP. This phenomenon prompts a critical investigation: should we prioritize reusing high-quality data or incorporate lower-quality yet more abundant data to escalate training dataset sizes?

To encapsulate this inherent trade-off between data scale and quality, we devise a hierarchical pyramid architecture for data pools. Given $m$ "top" OPs, we can create $2^m - 1$ combinations of these OPs, as depicted in the left-bottom area in Fig. 2. For example, the combination of three OPs, $\mathcal{OP}_{1,2,3}$, resides at the highest level of the hierarchy but results in the smallest data pool after Data-Juicer

data processing. The combinations of two OPs, such as $\mathcal{OP}_{1,2}$, are placed at a lower level, and the resulting data pool encompasses that of $\mathcal{OP}_{1,2,3}$ is several times larger in volume. Leveraging findings from Section 3.3.2, we can pinpoint preferred OP combinations within this pyramid structure. Progressing downward through the pyramid, data pools exhibit a descending average OP ranking (potentially indicative of reduced data quality) alongside an increase in volume.

To reconcile the desire for larger datasets without compromising quality, we devise two experimental strategies built upon this data pyramid: (1) iterative model training with data repetition from the top-layer (highest-quality) data pools, and (2) non-repetitive training incorporating progressively lower-quality, larger-volume data pools from the lower-layer pools. Specifically, we extract a predetermined quantity of data from the top-layer pool for iterative training with variable repetition rates. In parallel, as a baseline, we assemble a non-repeating dataset by consolidating pools from the upper to lower pyramid levels to match the size of the iterated dataset.

These comparative studies allow us to qualitatively assess the efficacy of data reuse compared to the inclusion of suboptimal data within a fixed dataset size. Notably, all training data is uniformly sampled from the corresponding data pools derived from $\mathcal{D}$, ensuring that the relative proportions of the pyramid data pools remain consistent as $\mathcal{D}$ expands. Consequently, the insights gained from these experiments can be extrapolated to larger-scale data contexts, informing efficient and effective scaling of data-model co-development practices.

## 4 PRACTICAL APPLICATIONS AND INSIGHTS

### 4.1 USE-CASES AND EXPERIMENTAL SETTINGS

In this section, we demonstrate the versatility of the Data-Juicer sandbox through two practical scenarios: image-to-text and text-to-video generation, illustrating its effectiveness in multimodal data-model co-development. These examples instantiate the behavior hooks and capability classes detailed in Sec. 3.2, following the probe-analyze-refine workflow outlined in Sec. 3.3.

In the subsections, we delve into the primary findings and insights, with complete results provided in Appendix C. Initially, we investigate the impact of single-OP data pools and summarize fundamental insights (Sec. 4.2). Building upon the identified "Top" OPs, we then explore different combinations (Sec. 4.3) and construct a data pyramid hierarchy to shape optimal data recipes (Sec. 4.4). Finally, leveraging the insights gained, we scale up the data and models to train superior models (Sec. 4.5).

Specifically, in the main page, we adopt the Mini-Gemini model for the image-to-text task, and the EasyAnimate and T2V-Turbo models for the text-to-video task. In Appendix C.7, we extend our experiments to the CLIP models and summarize all conducted experiments from different scale aspects in Appendix C.8. Our evaluation occurs on established benchmarks with over 30 metrics, employing more than 40 Data-Juicer OPs to gain in-depth insights into data-model co-development. We report the average performance compared to the baseline model trained on $\mathcal{D}_{\text{random}}$. More detailed configurations can be found in Appendix B, including the sources and sizes of data pools ($\mathcal{D}, \mathcal{D}_{\text{random}}, \mathcal{P}_i, \mathcal{P}_{\mathcal{S}}$), model performance metrics, and functionalities of studied OPs.

### 4.2 RANKING SINGLE-OPERATOR DATA POOLS

Table 1 illustrates the results of models trained on top-performing data pools alongside some of particular interest. For detailed experimental results of all examined OPs, please refer to Appendix C.1.

> **Observation 1 (Data vs. Model)**
>
> Generative models' efficacy is intimately tied to the fidelity of modalities they are trained to generate, which can be explicitly reflected in filtering processes on training data.

In text-to-video generation, video-related OPs occupy the top performance ranks, with the top three results all being video-related (0.96 for high *video aesthetics score*, 0.82 for low *video NSFW score*, and 0.79 for high *frames-text similarity*), presenting a notable gap over the fourth-ranked text-related OP (0.54 for low *special-characters ratio*). Conversely, in image-to-text generation, text-related statistics such as *text action number* and *language score* appear to be highly influential.

| Task | Rank | OP-Generated Statistics | Average Performance Changes (%) | | |
|------|------|-------------------------|--------------------------|---|---|
| | | | $\mathcal{P}_{i,low}$ | $\mathcal{P}_{i,middle}$ | $\mathcal{P}_{i,high}$ |
| Image-to-Text | 1 | *Image NSFW Score* | $7.13 \pm 4.29$ | $18.44 \pm 18.45$ | $\mathbf{66.38 \pm 32.65}$ |
| | 2 | *Text Action Number* | $\mathbf{59.90 \pm 46.49}$ | $0.29 \pm 2.16$ | $-2.05 \pm 2.48$ |
| | 3 | *Language Score* | $\mathbf{49.90 \pm 53.82}$ | $0.85 \pm 2.87$ | $-1.43 \pm 2.40$ |
| | 4 | *CLIP Image-Text Similarity* | $1.20 \pm 4.86$ | $-1.81 \pm 2.88$ | $\mathbf{49.81 \pm 44.72}$ |
| | 5 | *Phrase Grounding Recall* | $-0.49 \pm 3.87$ | $-0.58 \pm 6.12$ | $\mathbf{49.39 \pm 29.83}$ |
| | 15 | *Aesthetics Score* | $11.94 \pm 12.21$ | $\mathbf{16.58 \pm 25.70}$ | $0.16 \pm 3.67$ |
| Text-to-Video | 1 | *Video Aesthetics Score* | $-0.98 \pm 0.08$ | $0.13 \pm 0.09$ | $\mathbf{0.96 \pm 0.13}$ |
| | 2 | *Video NSFW Score* | $\mathbf{0.82 \pm 0.36}$ | $-0.05 \pm 0.07$ | $-0.57 \pm 0.07$ |
| | 3 | *Frames-Text Similarity* | $-1.45 \pm 0.69$ | $0.23 \pm 0.21$ | $\mathbf{0.79 \pm 0.15}$ |
| | 4 | *Special-Characters Ratio* | $\mathbf{0.54 \pm 0.36}$ | $-0.13 \pm 0.70$ | $-0.14 \pm 0.10$ |
| | 11 | *Text Action Number* | $0.18 \pm 0.56$ | $-0.71 \pm 0.28$ | $\mathbf{0.37 \pm 0.28}$ |
| | 13 | *Video Motion Score* | $-0.55 \pm 0.40$ | $\mathbf{0.33 \pm 0.21}$ | $0.32 \pm 0.15$ |
| | 18 | *Language Score* | $-0.21 \pm 0.01$ | $-0.03 \pm 0.38$ | $\mathbf{0.09 \pm 0.03}$ |

Table 1: The results for some investigated OPs gaining top performance with models trained on single-OP data pools, including their statistical dimensions and the average performance changes relative to the baselines trained on $\mathcal{D}_{random}$. Full ranking table can be found in Appendix C.1.

---

**Observation 2 (Diversity vs. Quality)**

Image-to-text models place greater emphasis on data diversity, whereas text-to-video models prioritize data quality.

---

A deeper analysis of Table 1 in terms of *NSFW scores* and *language scores* reveals that the studied image-to-text model requires more diverse data compared to the text-to-video model. Intuitively, high-scoring images or videos in terms of *NSFW* (Not Safe For Work) content are generally rare and occupy the long tail of the data distribution. Consequently, datasets with high *NSFW scores* tend to be more diverse. Similarly, texts with low *language scores* indicate that the language of the text is more difficult to identify, suggesting that such texts are less common and correspond to a more diverse data pool. Please refer to Appendix C.2 for further quantitative evidence and analysis.

---

**Observation 3 (Spatiotemporal Dynamics)**

Dynamic information in the data presents a heightened learning challenge for image-to-text generation compared to text-to-video generation.

---

Moreover, given the static nature of images, image-to-text generation sometimes requires parsing dynamic content, which compels models to engage in creative inference for accurate interpretation and response. This challenge is evident in the difficulties image-to-text models encounter when trained on data rich in dynamic information. As shown in Table 1, image-to-text models perform commendably with fewer *text action numbers*, while text-to-video models display contrasting trends regarding the values of *text action numbers* and *video motion scores*.

---

**Observation 4 (Modality Alignment)**

A high degree of alignment between different modalities within the data is crucial for model performance in both image-to-text and text-to-video generation.

---

Finally, both image-to-text and text-to-video models show a common preference for a higher degree of alignment between modalities. This is supported by their exemplary performance in scenarios where measures of *image-text similarity*, *phrase grounding recall*, and *frames-text similarity* are high.

### 4.3 SHAPING DATA RECIPES OF TOP-3 OPERATOR COMBINATIONS

Based on insights from single OP experiments, we can combine "Top" OPs into recipes for data filtering to obtain higher-quality data. Fig. 3 illustrates model performance changes trained on different recipes $\mathcal{P}_S$ compared to the models trained on $\mathcal{D}_{\text{random}}$.

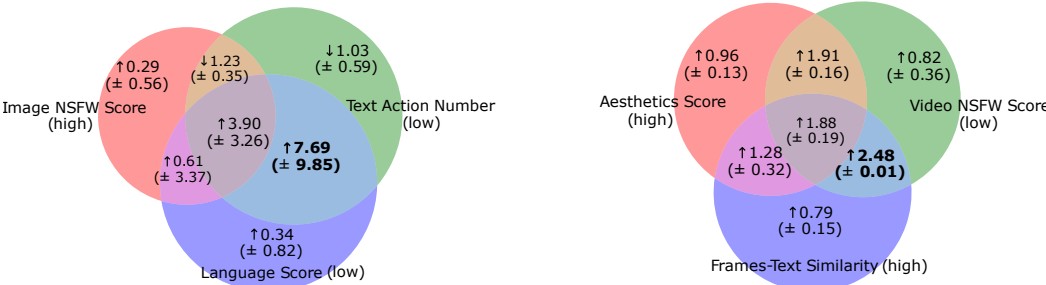

(a) Top-recipes for image-to-text model          (b) Top-recipes for text-to-video model

Figure 3: The model performance changes (%) from recipes combined with the top-3 overall OPs listed in Table 1. In image-to-text generation, the degree of performance change is smaller than in single-OP experiments due to the reduced data volume of $\mathcal{P}_S$. Details can be found in Appendix B.1.

> **Observation 5 (Effect of Sequential Combination)**
>
> The optimal data recipe does not necessarily arise from combining the best individual OPs, nor does adding more high-performing OPs always lead to superior outcomes.

Given the exponential growth in possible OP combinations, we focus on experimenting with combinations of three OPs at a time. An intuitive strategy is to combine the top three OPs with the best overall performance from Table 1. However, as shown in Figures 3(a) and 3(b), combining higher-performing OPs does not always yield better results. For instance, in the image-to-text experiment depicted in Fig. 3(a), the data pool with a high *image NSFW score*, despite performing best in single-OP experiments, generally diminishes performance when combined with others. Similarly, in Fig. 3(b), while pairwise combinations of OPs show positive gains, integrating the top-performing OP into the combination of high *frames-text similarity* and low *video NSFW score* reduces the relative improvement over the baseline from 2.48% to 1.88%.

Additionally, we categorize OPs based on correlations and select the optimal OP from each category to form recipes. However, detailed results in Appendix C.4 show that this approach also did not yield better outcomes, challenging the common assumption in existing SOTA works that stacking various intuitively useful data cleansing actions can synergistically enhance performance.

> **Observation 6 (Effect of Seed OPs)**
>
> The performance of a single OP is positively correlated with the performance of the recipe created from its combination. Starting with high-performing OPs is a good initial step in exploring optimal higher-order data recipes.

Although the Top-3 OP recipes exhibit suboptimal performance, we observe positive gains when combining some pairs of OPs in them, outperforming both single top-1 and top-3 combinations. For example, combining `TextActionFilter` and `LanguageIDScoreFilter` for the image-to-text generation, as well as `VideoNSFWFilter` and `VideoFramesTextSimilarityFilter` for the text-to-video generation, has proven to be highly effective.

### 4.4 HARNESSING MORE HIGH-QUALITY DATA

As discussed in Sec. 3.3.3, we can construct pyramid-shaped data pools by combining different "top" OPs and then explore when to reuse high-quality data and when to introduce suboptimal data to ensure diversity. Fig. 3 presents specific performance for $\mathcal{P}_S$ within this pyramid structure, from which we can select candidates to be duplicated. Specifically, for the image-to-text model, we choose

the data pool with low *text action number* and *language score*, and for the text-to-video model, we select the data pool with a low *video NSFW score* and a high *frames-text similarity*.

> **Observation 7 (Effect of Duplicates)**
>
> Duplicating high-quality data benefits both image-to-text and text-to-video models. For the image-to-text model, four repetitions of high-quality data yield the best empirical performance, while for the text-to-video model, a larger repetition ratio of six to ten times proves effective.

For the text-to-video model (Fig. 4(b)), we find that setting the expansion rate to 2 allows the model to achieve performance comparable to that obtained by training with the same data volume while incorporating data from the suboptimal pool. This indicates that data duplication does not negatively impact performance. The results further demonstrate that this trend continues up to six repetitions, with linear performance improvements. After six repetitions, some detrimental effects begin to emerge, although improvements are still noted even after ten repetitions, suggesting a relatively minor negative impact. In contrast, for the image-to-text model (Fig. 4(a)), repeating the data twice results in a 15% reduction in relative improvement compared to using double the amount from the suboptimal pool. Additionally, the benefits of reusing this data diminish with further repetitions, and negative effects become apparent after eight repeats.

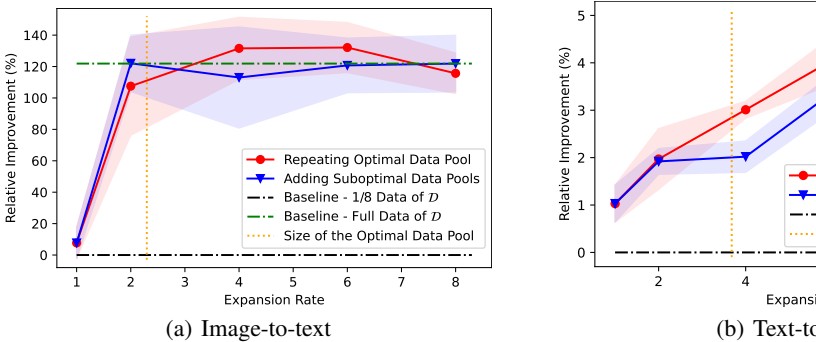

(a) Image-to-text            (b) Text-to-video

Figure 4: Relative improvement over the baseline for both image-to-text and text-to-video models is observed when these are trained on repeated top data pools and when additional suboptimal data pools are included. The pools are sourced from pyramid hierarchy in Fig. 3.

Furthermore, Fig. 4(b) shows that the introduction of suboptimal data significantly impacts performance compared to reusing high-quality data. At an expansion rate of 4, including suboptimal data results in a notable performance gap, even though it comprises only 8% of the data. Although expansion to 6 allows for some performance gain from data diversity, the overall performance declines with a higher proportion of suboptimal data.

Besides, experiments with the image-to-text model highlight the critical role of data quality, as shown in Fig. 4(a), where performance does not improve with the addition of suboptimal data. These findings suggest that while the text-to-video model shows greater tolerance for data diversity, both models underscore the importance of data quality, aligning with Observation 2 in Section 4.2.

### 4.5 APPLYING DATA-MODEL INSIGHTS IN LARGER-SCALE SCENARIOS

In this section, we apply the insights gained from our previous experiments to larger-scale models and datasets. Specifically, for the image-to-text generation task, we follow Observations 6 and 7, utilize the top data pool from Fig. 3(a) with 4x repetition, resulting in a pretraining dataset of 637k samples, which constitutes only half of its original pre-training dataset size. Then we train the MGM-2B model using this new pretraining dataset and its original full fine-tuning dataset.

Table 2 presents the experiment results. Since the official MGM-2B model has not been evaluated on MMBench-CN, we trained a reproduced version using the official training scripts, and its performance is closely aligned with the benchmarks reported for the official model. We can see that compared to the baseline trained on the full dataset, our Data-Juicer version—trained on only 1/10 of the distinct instances and 1/2 of the total instances—achieves superior performance. Notably, the only difference between these models is the pretraining data, which underscores the effectiveness of our insights.

| MGM-2B | Num. of Instances | Avg. Perf. Changes (%) | MMBench | MMBench-CN | MME-Perception | MME-Cognition |
|---|---|---|---|---|---|---|
| Baseline* | 1226k | - | 59.2 | 51.6 | 1334 | 302 |
| Data-Juicer | **159k (x4)** | **+2.12** | 62.08 | 52.32 | 1323.37 | 311.07 |

Table 2: Performance of MGM-2B with different pretraining datasets. MGM-2B pretrained with only 1/10 distinct instances and 1/2 of the total instances performs better. than the baseline trained on the full dataset. The "*" indicates our reproduced version that is comparable with the official version.

For the text-to-video task, in line with the best data recipe from Fig. 3 and Observation 6, we scale up the data pool on the full-size candidate video datasets introduced in Sec. B.2, with low *video NSFW score* and high *frame-text similarities*, encompassing approximately 375k instances totally. Additionally, following Observation 7, we conduct no more than six model training passes through this dataset. From a data development view, this transition allows us to probe the scalability of our methodologies, advancing from the 40k data pool used in Sec. 4 to a significantly more voluminous dataset (228k), approximately 5.7× larger. From a model development view, we undertake a stringent challenge to assess the transferability of our findings across different model architectures, by replacing the training model from the previous EasyAnimate with another SOTA model.

| Models (Ranked by leaderboard) | Board Avg. (%) | Uniform Avg. (%) | Quality Avg. (%) | Semantic Avg. (%) |
|---|---|---|---|---|
| 1. **Data-Juicer (DJ, 228k)** | **82.53** | **81.26** | 83.38 | **79.13** |
| **Data-Juicer (T2V, 147k)** | 82.10 | 80.54 | 83.14 | 77.93 |
| 2. Gen-3 (RunwayML, 2024) | 82.32 | 79.64 | **84.11** | 75.17 |
| 3. VEnhancer (VC2) (He et al., 2024a) | 81.97 | 80.00 | 83.27 | 76.73 |
| 4. Kling (2024-07) (Kuaishou, 2024) | 81.85 | 79.54 | 83.39 | 75.68 |
| 8. T2V-Turbo (VC2) (Li et al., 2024a) | 81.01 | 78.67 | 82.57 | 74.76 |

Table 3: Leading models on the VBench leaderboard as of Sep 23, 2024. "Board Avg." denotes the weighted average scores across 16 metrics defined by VBench and "Uniform Avg." denotes the arithmetic average. Full training details and ranking table are in Appendix C.5 and C.6 respectively.

Table 3 showcases our notable performance on the VBench leaderboard, achieving a new rank-one SOTA model. We first enhance T2V-Turbo (the last row) with data-enhanced distillation training on 147k instances (the second row), and then self-distill it with other 228k instances (the first row). Compared to our baseline, our method yields a notable uplift of 1.53% on the *Board Average* score, with the *Quality Average* score experiencing a boost of 0.82%, and a particularly pronounced elevation in the *Semantic Average* score, which escalates by 4.38%. This underscores the pivotal role played by our proposed data-model refinement strategies within this context. The `VideoFramesTextSimilarityFilter` effectively bolsters the alignment between the generated videos and the corresponding prompts, while the `VideoNSFWFilter` safeguards the maintenance of high video generation quality standards. It is also worth highlighting that our approach charts a novel trajectory for the advancement of text-to-video models. While both T2V-Turbo and VEnhancer embody architectural refinements building upon the foundation of VideoCrafter-2.0 (VC2), our method augments T2V-Turbo's performance through a synergistic data-model co-development workflow. For detailed experimental setups and supplementary ablation studies regarding T2V-Turbo, readers are directed to Appendix C.5.

## 5 CONCLUSIONS

In this paper, we introduced the Data-Juicer Sandbox, a pioneering open-source suite designed to facilitate the co-development of multimodal data and generative models. By integrating a flexible and comprehensive array of customizable components at different levels, our sandbox enables systematic, cost-effective exploration and optimization, effectively bridging often-disjoint domains of data and model development. Through applying the proposed "Probe-Analyze-Refine" workflow, we showcased how our sandbox can yield not only significant improvements in both dataset and models, but also valuable insights into the complex interplay between data processing and model performance.

## REPRODUCIBILITY STATEMENT

Reproducibility is essential for validating research outcomes. To facilitate this, we have organized detailed description within the appendix of our paper. Key components of our experimental setup, including implementation details for the image-to-text and text-to-video use cases can be found in Appendix B.1 and Appendix B.2 respectively. We provide details into the methodologies for combining multiple operators based on their correlations in Appendix B.5, as well as comprehensive descriptions of model training configurations (Appendix B.6) and performance metrics (Appendix B.7). Furthermore, Appendix B.8 outlines the functionalities and statistics of the Data-Juicer OPs utilized in our experiments.

All codes, datasets, and models of our work are openly accessible and actively maintained at the anonymous link.

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

APPENDIX

Our appendix is organized as follows:

- In Appendix A, we present more detailed introduction to related works.

- In Appendix B, we present **details about the sandbox implementation and experimental setup**, including the image-to-text use case (Appendix B.1), the text-to-video use case (Appendix B.2), the two strategies used to combine multi-OPs into recipes based on their correlations (Appendix B.5), the model training configurations (Appendix B.6), the adopted model performance metrics (Appendix B.7), and the functionalities and corresponding statistics of the studied Data-Juicer OPs (Appendix B.8).

- In Appendix C, we present more **experimental results**, including the complete operator ranking (Appendix C.1), the data diversity analysis (Appendix C.2), the correlation analysis (Appendix C.4), model performance on recipes derived from the correlation analysis (Appendix B.5), a detailed performance and ablation study of the Data-Juicer T2V models (Appendix C.5), and the detailed V-Bench leaderboard results (Appendix C.6). We also extended our experiments to the image-text foundation model pretraining task in Appendix C.7. Finally, we provided an overview of our experiments in Appendix C.8.

## A  DISCUSSION ON MORE RELATED WORKS

**Model-Centric Progress in Multimodal Large Models.** Multimodal large models have captivated researchers with their formidable capabilities (OpenAI, 2024a;b), leading to a surge in model-centric development efforts. These focuses mainly lie in refining training algorithms (Caffagni et al., 2024; Li et al., 2024a; Zhang et al., 2024), advancing model architectures and components (He et al., 2024a; Yin et al., 2024; Jiao et al., 2024), and harnessing the models' potential for various applications (Wang et al., 2024b; Liu et al., 2024; Zhou et al., 2024). There is a growing consensus that transformer-based scaling is predominant (Xu et al., 2023). However, the high computational requirements imposed by scaling laws (Xu et al., 2024c) and the optimization challenges inherent to large models (Manduchi et al., 2024) often confine insights to specific datasets or vague data characteristics. This situation leaves a significant gap in comprehending the extent to which models' performance and behavior hinge upon implicit assumptions and inductive biases embedded within the underlying data distributions. In contrast, our work demonstrates a feasible and promising path to fill in this gap by explicitly linking data processing effects with the downstream performance of trained models through numerous contrastive sandbox experiments.

**Trends in Data-Centric Development for Multimodal Large Models.** An emerging trend shifts the focal point from models to data (Jakubik et al., 2024; Bai et al., 2024), underscored by the notion that large models function akin to data compressors (Delétang et al., 2024). Echoing the principle of "garbage in, garbage out", meticulous data processing is recognized as pivotal. Efforts now isolate data manipulation as a primary experimental variable in multimodal generative modeling (He et al., 2024b). Nonetheless, multimodal data processing involves highly heterogeneous processing workflows, vast quantities, diverse types, and the high cost of training downstream models. This complexity results in predominantly heuristic approaches, such as data filtering and synthesis guided by human intuition (Long et al., 2024).

For example, one well-studied model type is CLIP. DataComp (Gadre et al., 2023) introduces a benchmark to filter out high-quality data from 12.8 billion image-text pairs in Common Crawl to train better CLIP models, considering Filter Operators such as CLIP score, image size, and caption length. MetaClip (Xu et al., 2024a) aims to reproduce CLIP's data by introducing a raw data pool and finding a balanced subset based on CLIP's concepts. Unlike these data-centric approaches that isolate the model and training settings, focusing solely on the quality and scale of training datasets, our work emphasizes systematic methodologies for data-model co-development, considering both data and models equally important. Specifically, we incorporate performance signals from sandbox reference models on many downstream tasks, conducting importance and correlation analysis to link data pools and these model metrics. Additionally, we explored more model types beyond CLIP, such as LLaVA-like and DiT-based models for image-to-text and text-to-video tasks, and identified better training datasets for these models using our workflow.

**Open-Source Infrastructure for Multimodal Large Model Development.** The landscape for multimodal model development has advanced significantly, boasting a variety of strong infrastructures and frameworks for training and evaluation. Prominent examples include Transformers (Wolf et al., 2020), Diffusers (von Platen et al., 2022), NeMo (Harper et al.), MMagic (MMagic Contributors, 2023), ESPNet (Peng et al., 2023), and MMBench (Liu et al., 2023).

However, when it comes to multimodal data infrastructure, the primary contributions have been datasets and dataset-specific preprocessing tools, such as DatasetsHub from HuggingFace (Lhoest et al., 2021). The standardization and efficient utilization of practical expertise and foundational data processing capabilities remain unaddressed. Existing frameworks for data processing predominantly focus on single-modal data (TogetherAI, 2023; Bradski, 2000; Hwang et al., 2023), underlining the early stages of development in systematic platforms for multimodal data (Chen et al., 2024a; Du et al., 2023).

Recognizing the critical interplay between datasets and models—where comprehensive, high-quality datasets enhance model performance, and advanced models contribute to the generation of even more refined datasets—our work stands out by introducing an innovative intermediary layer. We seamlessly integrate cutting-edge model-centric multimodal infrastructure with the Data-Juicer system. This integration fosters a streamlined and insightful co-development environment for both models and data, bridging the current gap and setting a standard for future efforts in the multimodal domain.

## B    DETAILS OF SANDBOX APPLICATION

In this section, we present the implementation and experiment details about the two use cases in 4, which illustrate the applications of the Data-Juicer sandbox and demonstrate its versatility and effectiveness in enhancing multimodal data-model co-development.

### B.1    IMAGE-TO-TEXT GENERATION

Our first task focuses on foundational image understanding ability, by experimenting on Mini-Gemini (MGM-2B), a state-of-the-art (SOTA) 2 billion parameter multimodal LLM (Li et al., 2024b). The training protocol for MGM-2B involves two stages: pretraining and fine-tuning. Our experimental focus lies in the pretraining phase, which seeks to harmonize visual and textual representations. We utilize the original pretraining dataset as our original dataset $\mathcal{D}$, consisting of approximately 1.2M instances. We set the size of $\mathcal{D}_{sample}$ as 200k. The single-OP data pools $\mathcal{D}_i$ and multi-OP data pools $\mathcal{D}_{\mathcal{S}}$ are capped at a maximum of 200k instances, ensuring consistency of data pool size. To match the down-sampling rate used during pretraining, the fine-tuning dataset is sampled into a 240k instance subset.

We first conduct single-OP experiments (Section 4.2) that encompasses 22 text-image relevant OPs from Data-Juicer, split evenly between text-only and image-related multimodal OPs. After the two-stage training, model evaluation is conducted on established benchmarks including TextVQA (Singh et al., 2019), MMBench (Liu et al., 2023), and MME (Fu et al., 2023).

For multi-OP data pools (Section 4.3), we identify the top-3 highest-performing OPs from single-OP experiments and study their possible combinations. Additionally, we analyze the correlations among the 23 data statistics produced by 22 OPs capable of generating instance-level stats [1]. Employing a hierarchical clustering algorithm  (Ward Jr, 1963), these OPs are grouped into three clusters based on correlation coefficients, with the highest-performing OP from each cluster selected for combination testing. To ensure a robust and fair comparison, we must acknowledge the constraints imposed by the limited data volume within the highest-tier data pool. As the number of combinations increases, the available dataset size diminishes. In particular, the size of $\mathcal{P}_{\mathcal{S}}$ was reduced from 200k samples to 159k samples during the Top-3 combination experiments. Similarly, in the cluster-wise combination experiments, the dataset size decreased from 200k samples to 126k samples.

Next, in Section 4.4, we explore the optimal OP combination based on previous experiments and adopt the methodology from Section 3.3.3 for comparative experiments on training with repeated data versus non-repeated data. Note that due to filtering, the final instance count decreases from 200k

---

[1]The image height and width are produced by one OP.

to approximately 159k after the OP combination. These data are then repeated in increments from double to eightfold, mirroring the size of the original pretraining set.

Collectively, all these experiments yield profound insights into image-to-text model training, data processing, and iteration strategies from a data-model co-development perspective, further verified in the larger-scale scenario in Section 4.5.

## B.2 Text-to-Video Generation

For the second task, text-to-video generation, we adopt the advanced DiT-based models, EasyAnimate (Xu et al., 2024b), which originally integrates diverse datasets totaling 1.2M instances from InternVid (Wang et al., 2023b) (606k), Panda-70M (Chen et al., 2024c) (605k), and MSR-VTT (Xu et al., 2016) (6k). The studied baseline model is trained on a subset of 40k instances, employing LoRA (Hu et al., 2021) for efficiency. As a result, the size of $\mathcal{D}$ is 1.2M, and the size of $\mathcal{D}_{sample}$, the single-OP data pools $\mathcal{D}_i$ and multi-OP data pools $\mathcal{D}_{\mathcal{S}}$ are all 40k. Model outputs are assessed using VBench (Huang et al., 2024) across 16 metrics on video quality and video-text matchness.

Our investigation covered 21 OPs, including 13 text-only OPs and 10 video-related multimodal OPs. Analogous to the image-to-text generation, we conduct single-OP and multi-OP combination experiments, in Section 4.2 and Section 4.3 respectively. However, given the reduced relevance of data statistics in video-related OPs, our analysis centers on the correlations among the 16 VBench evaluation metrics. These metrics are clustered into three groups, with the best-performing OP selected from each group.

Through OP combination experiments, we pinpoint the most effective set of OPs. In Section 4.4, we then sample 40k instances from the filtered data pool and repeat the training process for up to 10 epochs. For comparative analysis, we adhere to the method outlined in Section 3.3.3, selecting larger data volumes (80k, 120k, ..., up to 400k instances) for single-epoch training. To examine the effectiveness of the derived insights for text-to-video data-model co-development, we finally incorporate them into larger-scale scenarios in Section 4.5.

## B.3 Discussion on the Determination of Data Pools

In the sandbox experiments in Section 4, we split the data pools into three buckets. The number of buckets reflects the trade-off between data intervention intensity (via operator stats) and the reliability of model feedback (the metric changes $\Delta$ of post-training downstream tasks compared to the models trained on random sampled data pools). More buckets lead to greater statistical differences between buckets with different ranks (especially the first and last ones), strengthening attribution to data processing effectiveness. However, more buckets also reduce per-bucket data, increasing the risk of inadequate data for models to exhibit reasonable $\Delta$.

As a result, we do not aim for models to be "training done" or "converged" in this sandbox experiment setting. Instead, we want to observe enlightening changes—positive or negative—in models after targeted data intervention versus random data sampling. In our early experiments, we tested bucket counts of [2,3,4,5] to evaluate whether a model trained on randomly sampled data could reasonably decrease loss after one epoch and show statistically significant changes on downstream tasks. Our findings indicate that three buckets are empirically good for our scenarios. Once determined, all controlled experiments are aligned to one complete epoch and matched to the random pool data size.

## B.4 Extensibility of the Sandbox Suite

As a middleware, the sandbox itself does not impose any additional specific hardware dependencies. Instead, it inherits the dependencies of the integrated underlying libraries/frameworks. Besides, to simplify dependencies and avoid redundancy in an "all-in-one" environment, we have introduced and employed a lazy-loader mechanism at the Python package level.

Besides, the utilized Data-Juicer's operators are not limited to simple data filters; they now encompass a wide range of functionalities, including 100+ Mappers, Filters, Deduplicators, and Selectors. This diversity allows for research on various types of data processing utilities within the Sandbox. For

example, Mappers can be employed to examine the effects of data augmentation and editing on downstream model task performance. We reserved further exploration beyond filters for future work.

### B.5 STRATEGIES TO COMBINE OPS ACCORDING TO CORRELATIONS

In addition to assembling the OPs with the overall best performance, we also incorporate an analysis of inter-OP relationships into our recipe formulation process. Our workflow accommodates two strategies, with specific applications detailed in Section C.3:

- The first method involves computing Pearson correlation coefficients between the statistics generated by these OPs. Using a hierarchical clustering algorithm (Ward Jr, 1963), we group the OPs into $k$ clusters. From each cluster, we select the OP whose data pool yields the strongest model performance to form potential combinations.

- Alternatively, leveraging the outcomes of single-OP tests, we calculate Pearson correlation coefficients for each pair of dimensions within the evaluation metrics. Hierarchical clustering is again employed to categorize the metrics into $k$ classes. The top-performing OP from each class is chosen to create the combinations. This approach allows us to investigate whether these combinations lead to concurrent improvements or mutual inhibition across the evaluative metrics.

### B.6 TRAINING CONFIGURATIONS

For the image-to-text generation, we conducted experiments on one of the state-of-the-art MLLMs, Mini-Gemini-2B (Li et al., 2024b). We train the whole model from scratch with less training data (about 1/6 of the original training datasets) in baseline experiments to make sure each experiment can be finished within one day. We keep every training setting (e.g. learning rate scheduler, global batch size) the same as the original model except for training datasets and training devices. For single-OP and OP combination experiments are trained on only 1 A100 GPU for each experiment so we increase the number of gradient accumulation steps from 4 to 32 to keep the same global batch size. For experiments of duplicating high-quality datasets, 8 A100 GPUs are involved to train the model, and the number of gradient accumulation steps is restored to 4. Each experiment is repeated 3 times with different random seeds to make the final results more reliable.

For text-to-video generation, we adopt the advanced DiT-based EasyAnimate (Xu et al., 2024b) model, which integrates diverse datasets totaling 1.2M instances from InternVid (Wang et al., 2023b) (606k), Panda-70M (Chen et al., 2024c) (605k), and MSR-VTT (Xu et al., 2016) (6k). Baseline experiments are executed on a subset of 40k instances, employing LoRA (Hu et al., 2021) for efficiency. During training, we maintain a video resolution of 256x256, sample every other frame, and randomly select sequences of 16 consecutive frames. The training process involves performing a backward pass for the loss of every 8 samples, with single-OP and OP combination experiments trained on a single GPU with a batch size of 8 for 5k steps, amounting to approximately 16 GPU hours per training run. Experiments for duplicating high-quality data, as well as larger-scale training, are conducted with a batch size of 1 across 8 GPUs. The models employ the Adam optimizer for training, with a learning rate set to $2 \times 10^{-5}$, weight decay parameter at $3 \times 10^{-2}$, and epsilon configured to $10^{-10}$. Each experiment is repeated twice with random seeds of 42 and 45, respectively.

### B.7 PERFORMANCE METRICS

In the paper, we mainly report overall performance as the relative changes over the baseline in terms of the average across all model metrics with normalization as follows:

$$\frac{\sum_i^N s_i/N - \sum_i^N s'_i/N}{\sum_i^N s'_i/N} = \frac{\sum_i^N (s_i - s'_i)}{\sum_i^N s'_i}, \tag{1}$$

where $N$ is the number of involved metrics, $s_i$ is the score of $i$-th model measurement metric, $s'_i$ is the corresponding score gained by the baseline model trained on randomly sampled data. Below are the specific evaluation metrics involved in this study.

**TextVQA, MMBench, MME.** These benchmarks serve as critical evaluators of MLLM's proficiency in understanding images. TextVQA (Singh et al., 2019) specifically targets the assessment of MLLMs' abilities to read and reason about textual content embedded within images. MMBench (Liu

et al., 2023), a vast multimodal benchmark, encompasses perception and reasoning skills through a plethora of multi-choice questions, numbering in the thousands. Additionally, a Chinese translation, MMBench-CN, is integrated for broader accessibility. MME (Fu et al., 2023) focuses on the perceptual and cognitive competencies of MLLMs, incorporating 14 finely categorized subtasks, each addressing Yes/No inquiries underpinned by meticulously crafted guidelines.

**VBench.** We engage with VBench (Huang et al., 2024), a holistic benchmark suite tailored for the rigorous evaluation of video generative models. It facilitates granular and objective assessment across a spectrum of dimensions, deconstructing the concept of "video generation quality" into 16 discrete metrics. Each metric is assessed using a carefully curated suite of prompts, comprising 946 unique prompts, with the requirement to produce 5 videos per prompt.

Owing to the disparity in evaluation criteria and the inherent variability across different modalities, we discern that the magnitude of performance fluctuation in the image-to-text generation substantially exceeds that observed in the text-to-video generation. This discrepancy underscores again the need for nuanced data-model co-development in addressing the complexities inherent in each modality.

Across all experiments, results are reported as averages with standard deviations from 2 to 5 repetitions for the image-to-text and text-to-video tasks, respectively, due to their differing levels of variance.

### B.8 OPERATOR DESCRIPTIONS

The study involves 22 OPs for the image-to-text task and 21 OPs for the text-to-video task, comprising a distinct set of 31 OPs from Data-Juicer (Chen et al., 2024a). Their corresponding statistics and detailed descriptions are provided in Table 4.

## C ADDITIONAL EXPERIMENTAL RESULTS

### C.1 COMPLETE OPERATOR RANKING

In Table 5, we present complete numeric results conducted on individual OP experiments (Section 4.2), from which we can discern some more detailed observations.

In image-to-text generation, it is preferable for the input of training images to align as closely as possible with the original configuration of the vision tower, such as training dimensions (height, width, and sizes). Additionally, CLIP similarity scores tend to be more reliable than BLIP similarity scores. The BLIP similarity does not show much distinction and paradoxically, a lower similarity often results in better performance, which defies common sense. Images with excessively high aesthetic quality may offer limited assistance in feature alignment, while watermarks might have certain impacts on the OCR performance of the model.

In text-to-video generation, having a consistent aspect ratio for the training data is better than having ratios that are inconsistent but close to the 1:1 ratio used during training. For instance, a data pool with a 'middle' video aspect ratio consistently at 16:9 performs optimally. Videos with high video aesthetics scores and low video NSFW scores, as well as those with low video OCR-area ratios and high video motion scores, tend to be of higher quality. While single-text-related operators might not be as critical in text-to-video generation, they can still effectively filter out some dirty data.

### C.2 DIVERSITY ANALYSIS

In this subsection, we delve into the diversity of the data in interested data pools. Here, we confine our focus to statistical analysis of words within text data and compute their entropy. We operate under the assumption that the texts provide an accurate description of the images and videos. Consequently, the diversity inferred from the texts also serves as a proxy for the diversity of the associated images and videos.

The Table 6 shows the entropy of text words for different data pools, which can be normalized as follows:

$$\sum_w -\mathcal{P}(w) \log \mathcal{P}(w), \tag{2}$$

| OP Name | Modality | Statistics | Description |
|---|---|---|---|
| alphanumeric_filter | text | *Alphanumeric Ratio* | Alphanumeric ratio in the text. |
| character _repetition_filter | text | *Character Repetition Ratio* | Char-level n-gram repetition ratio in text. |
| flagged_words_filter | text | *Flagged Word Ratio* | Flagged-word ratio in the text |
| image_aesthetics _filter | image | *Image Aesthetics Score* | Aesthetics score of the image |
| image_aspect_ratio _filter | image | *Image Aspect Ratio* | Aspect ratio of the image |
| image_nsfw_filter | image | *Image NSFW Score* | NSFW score of the image |
| image_shape_filter | image | *Image Width/Height* | Width and height of the image |
| image_size_filter | image | *Image Size* | Size in bytes of the image |
| image_text_matching _filter | text, image | *BLIP Image-Text Similarity* | Image-text classification matching score based on a BLIP model |
| image_text _similarity_filter | text, image | *CLIP Image-Text Similarity* | Image-text feature cosine similarity based on a CLIP model |
| image_watermark _filter | image | *Image Watermark Score* | Predicted watermark score of the image based on an image classification model |
| language_id_score _filter | text | *Language Score* | Predicted confidence score of the specified language |
| perplexity_filter | text | *Text Perplexity* | Perplexity score of the text |
| phrase_grounding _recall_filter | text, image | *Phrase Grounding Recall* | Locating recall of phrases extracted from text in the image |
| special_characters _filter | text | *Special Character Ratio* | Special character ratio in the text |
| stopwords_filter | text | *Stopword Ratio* | Stopword ratio in the text |
| text_action_filter | text | Text Action Number | Number of actions in the text |
| text_entity _dependency_filter | text | *Entity Dependency Number* | Number of dependency edges for an entity in the dependency tree of the text |
| text_length_filter | text | *Text Length* | Length of the text |
| token_num_filter | text | *Token Number* | Token number of the text |
| video_aesthetics _filter | video | *Video Aesthetics Score* | Aesthetics score of sampled frames in the video |
| video_aspect_ratio _filter | video | *Video Aspect Ratio* | Aspect ratio of the video |
| video_duration _filter | video | *Video Duration* | Duration of the video |
| video_frames_text _similarity_filter | text, video | *Frames-Text Similarity* | Similarities between sampled frames and text based on a CLIP/BLIP model |
| video_motion_score _filter | video | *Video Motion Score* | Motion score of the video |
| video_nsfw_filter | video | *Video NSFW Score* | NSFW score of the video |
| video_ocr_area_ratio _filter | video | *Video OCR-Area Ratio* | Detected text area ratio for sampled frames in the video |
| video_resolution _filter | video | *Video Width/Height* | Width and height of the video |
| video_watermark _filter | video | *Video Watermark Score* | Predicted watermark score of the sampled frames in the video based on an image classification model |
| words_num_filter | text | *Word Number* | Number of words in the text |
| word_repetition _filter | text | *Word Repetition Ratio* | Word-level n-gram repetition ratio in the text |

Table 4: Overview of involved OPs in the study, including the modality they pertain to, along with their statistical data and detailed descriptions of these statistics.

where $w$ is a word and $\mathcal{P}$ is the distribution of words in a data pool. As we can see, data pools with higher *NSFW scores* and lower *language scores* have higher word entropy, suggesting greater diversity within these data pools.

| Task | OP-Generated Statistics | Average Performance Changes (%) | | |
|------|------------------------|------------------|------------------|------------------|
| | | Data Pool (Low) | Data Pool (Mid) | Data Pool (High) |
| Image-to-Text | *Image NSFW Score* | $7.13 \pm 4.29$ | $18.44 \pm 18.45$ | $\mathbf{66.38} \pm \mathbf{32.65}$ |
| | *Text Action Number* | $\mathbf{59.90} \pm \mathbf{46.49}$ | $0.29 \pm 2.16$ | $-2.05 \pm 2.48$ |
| | *Language Score* | $\mathbf{49.90} \pm \mathbf{53.82}$ | $0.85 \pm 2.87$ | $-1.43 \pm 2.40$ |
| | *CLIP Image-Text Similarity* | $1.20 \pm 4.86$ | $-1.81 \pm 2.88$ | $\mathbf{49.81} \pm \mathbf{44.72}$ |
| | *Phrase Grounding Recall* | $-0.49 \pm 3.87$ | $-0.58 \pm 6.12$ | $\mathbf{49.39} \pm \mathbf{29.83}$ |
| | *Image Width* | $\mathbf{42.04} \pm \mathbf{57.27}$ | $10.31 \pm 12.59$ | $1.35 \pm 4.36$ |
| | *Special Character Ratio* | $-3.08 \pm 0.63$ | $-0.75 \pm 1.61$ | $\mathbf{39.67} \pm \mathbf{58.82}$ |
| | *Flagged Word Ratio* | $\mathbf{38.48} \pm \mathbf{27.76}$ | $-0.39 \pm 0.43$ | $22.49 \pm 29.81$ |
| | *Image Height* | $\mathbf{35.66} \pm \mathbf{48.62}$ | $12.91 \pm 10.42$ | $18.73 \pm 27.32$ |
| | *Word Repetition Ratio* | $\mathbf{33.14} \pm \mathbf{23.39}$ | $2.59 \pm 5.31$ | $-0.55 \pm 2.90$ |
| | *Text Length* | $\mathbf{30.67} \pm \mathbf{28.54}$ | $-0.44 \pm 0.73$ | $-3.71 \pm 0.39$ |
| | *Stopword Ratio* | $3.34 \pm 5.05$ | $\mathbf{24.62} \pm \mathbf{36.73}$ | $-1.56 \pm 1.59$ |
| | *Image Size* | $0.76 \pm 0.55$ | $\mathbf{19.16} \pm \mathbf{27.29}$ | $1.58 \pm 2.20$ |
| | *Text Perplexity* | $-1.69 \pm 1.30$ | $16.70 \pm 24.49$ | $\mathbf{18.26} \pm \mathbf{23.02}$ |
| | *Image Aesthetics Score* | $11.94 \pm 12.21$ | $\mathbf{16.58} \pm \mathbf{25.70}$ | $0.16 \pm 3.67$ |
| | *Word Number* | $\mathbf{15.96} \pm \mathbf{29.01}$ | $-2.48 \pm 0.26$ | $-1.97 \pm 2.05$ |
| | *BLIP Image-Text Similarity* | $\mathbf{11.76} \pm \mathbf{22.83}$ | $1.74 \pm 2.49$ | $1.34 \pm 2.21$ |
| | *Image Watermark Score* | $-1.50 \pm 2.41$ | $7.51 \pm 12.82$ | $\mathbf{11.54} \pm \mathbf{13.14}$ |
| | *Alphanumeric Ratio* | $2.35 \pm 7.63$ | $-0.66 \pm 0.69$ | $\mathbf{8.71} \pm \mathbf{12.87}$ |
| | *Character Repetition Ratio* | $0.00 \pm 1.13$ | $-1.42 \pm 0.60$ | $\mathbf{7.94} \pm \mathbf{14.63}$ |
| | *Entity Dependency Number* | $1.35 \pm 1.81$ | $-0.87 \pm 1.15$ | $\mathbf{6.67} \pm \mathbf{8.44}$ |
| | *Token Number* | $\mathbf{6.31} \pm \mathbf{7.86}$ | $0.80 \pm 0.92$ | $0.33 \pm 6.45$ |
| | *Image Aspect Ratio* | $0.00 \pm 1.34$ | $\mathbf{1.89} \pm \mathbf{2.71}$ | $-0.02 \pm 1.12$ |
| Text-to-Video | *Video Aesthetics Score* | $-0.98 \pm 0.08$ | $0.13 \pm 0.09$ | $\mathbf{0.96} \pm \mathbf{0.13}$ |
| | *Video NSFW Score* | $\mathbf{0.82} \pm \mathbf{0.36}$ | $-0.05 \pm 0.07$ | $-0.57 \pm 0.07$ |
| | *Frames-Text Similarity* | $-1.45 \pm 0.69$ | $0.23 \pm 0.21$ | $\mathbf{0.79} \pm \mathbf{0.15}$ |
| | *Special-Characters Ratio* | $\mathbf{0.54} \pm \mathbf{0.36}$ | $-0.13 \pm 0.70$ | $-0.14 \pm 0.10$ |
| | *Token Number* | $\mathbf{0.53} \pm \mathbf{0.04}$ | $0.18 \pm 0.32$ | $0.41 \pm 0.25$ |
| | *Character Repetition Ratio* | $-0.29 \pm 0.27$ | $\mathbf{0.47} \pm \mathbf{0.80}$ | $0.18 \pm -0.52$ |
| | *Video Height* | $-0.10 \pm 0.21$ | $0.12 \pm 0.13$ | $\mathbf{0.46} \pm \mathbf{0.44}$ |
| | *Video OCR-Area Ratio* | $\mathbf{0.44} \pm \mathbf{0.04}$ | $0.02 \pm 0.63$ | $-0.66 \pm 0.23$ |
| | *Word Number* | $-0.49 \pm 0.07$ | $-0.41 \pm 0.72$ | $\mathbf{0.44} \pm \mathbf{0.45}$ |
| | *Entity Dependency Number* | $\mathbf{0.40} \pm \mathbf{0.01}$ | $0.28 \pm 0.48$ | $-0.18 \pm 0.44$ |
| | *Text Action Number* | $0.18 \pm 0.56$ | $-0.71 \pm 0.28$ | $\mathbf{0.37} \pm \mathbf{0.28}$ |
| | *Alphanumeric Ratio* | $-0.10 \pm 0.19$ | $0.20 \pm 0.19$ | $\mathbf{0.33} \pm \mathbf{0.17}$ |
| | *Video Motion Score* | $-0.55 \pm 0.40$ | $\mathbf{0.33} \pm \mathbf{0.21}$ | $0.32 \pm 0.15$ |
| | *Video Watermark Score* | $-0.27 \pm 0.27$ | $-0.25 \pm 0.25$ | $\mathbf{0.29} \pm \mathbf{0.16}$ |
| | *Text Perplexity* | $\mathbf{0.15} \pm \mathbf{0.69}$ | $-0.13 \pm 0.27$ | $0.09 \pm 0.56$ |
| | *Stopword Ratio* | $-0.01 \pm 0.05$ | $-0.48 \pm 0.22$ | $\mathbf{0.12} \pm \mathbf{0.07}$ |
| | *Video Aspect Ratio* | $-0.32 \pm 0.14$ | $\mathbf{0.11} \pm \mathbf{0.18}$ | $-0.02 \pm 0.40$ |
| | *Language Score* | $-0.21 \pm 0.01$ | $-0.03 \pm 0.38$ | $\mathbf{0.09} \pm \mathbf{0.03}$ |
| | *Word Repetition Ratio* | $0.00 \pm 0.17$ | $\mathbf{0.06} \pm \mathbf{0.24}$ | $-0.43 \pm 0.24$ |
| | *Video Duration* | $-0.58 \pm 0.05$ | $-0.16 \pm 0.09$ | $\mathbf{0.04} \pm \mathbf{0.84}$ |
| | *Text Length* | $-0.09 \pm 0.63$ | $-0.66 \pm 0.08$ | $\mathbf{0.03} \pm \mathbf{0.22}$ |

Table 5: The complete OP ranking, including their statistical dimensions and the improvements relative to the baseline. We consider three splits with low, middle, and high statistical values for each OP. The baseline used is based on random sampling with equal data volume.

## C.3 CORRELATION ANALYSIS

To investigate the intrinsic relationships between OPs and to support our recipe formulation, we explore relevance from the following two perspectives.

| Task | OP-Generated Statistics | Word Entropy | | |
|------|------------------------|--------------|--------------|---------------|
| | | Data Pool (Low) | Data Pool (Mid) | Data Pool (High) |
| Image-to-Text | *Image NSFW Score* | 6.97 | **7.35** | 7.29 |
| | *Language Score* | **7.47** | 7.32 | 6.98 |
| Text-to-Video | *Video NSFW Score* | 5.84 | **6.03** | 6.01 |
| | *Language Score* | **6.30** | 5.85 | 5.73 |

Table 6: The entropy of text words for data pools with different levels of *Image NSFW score* and *Language score*.

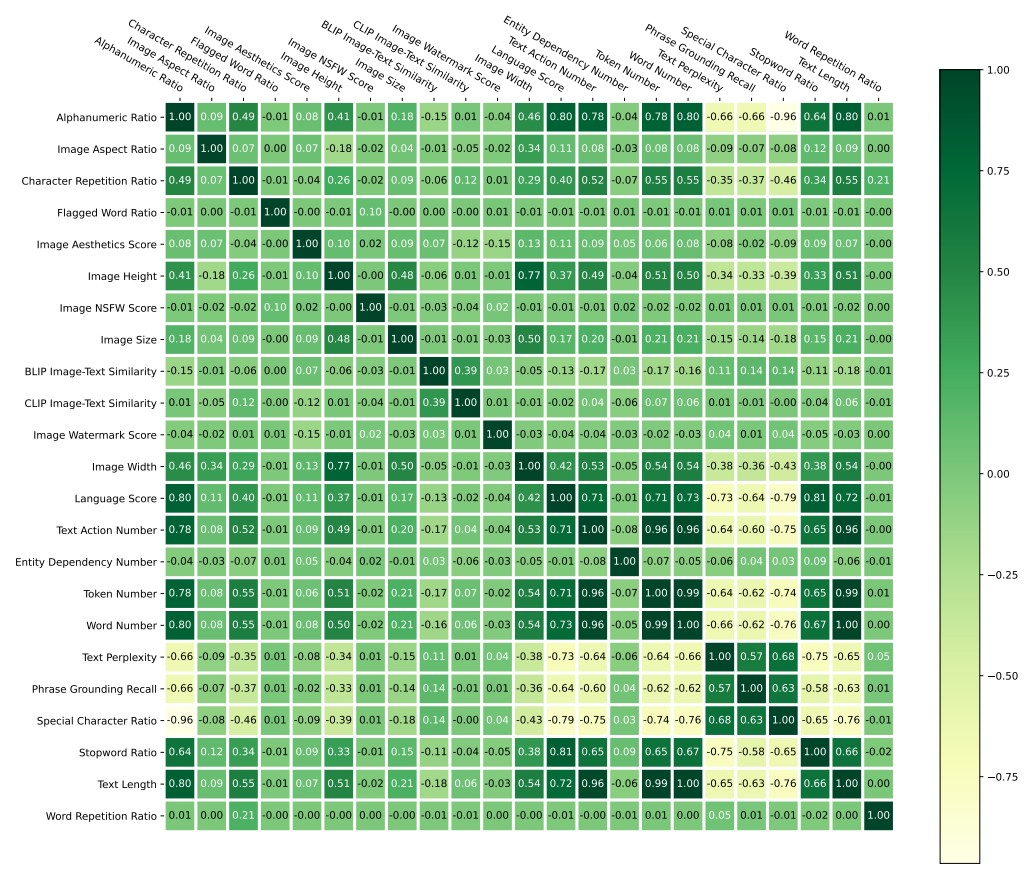

Figure 5: Pearson correlation coefficients for OP statistics in image-to-text generation.

First, we adopt the most direct approach by examining the Pearson correlation coefficients between the statistics of OPs, as illustrated in Figures 5 and 6. Intuitively, the associations between the statistics of OPs utilized in image-to-text generation appear to be significantly stronger than those in text-to-video generation. For instance, in image-to-text generation, phrase grounding recall shows a strong positive correlation with text perplexity and special character ratio, while exhibiting a strong negative correlation with the alphanumeric ratio, language score, number of text actions, stopword ratio, and text length. In contrast, in text-to-video generation, we observe relationships primarily among the purely textual OPs, while video-related operators are largely orthogonal to others. Therefore, for image-to-text generation, we categorize OPs into three groups based on the correlation of their statistics and select the optimal OP from each category to create combinations. However, this method does not appear appropriate for text-to-video generation due to the sparser correlations observed.

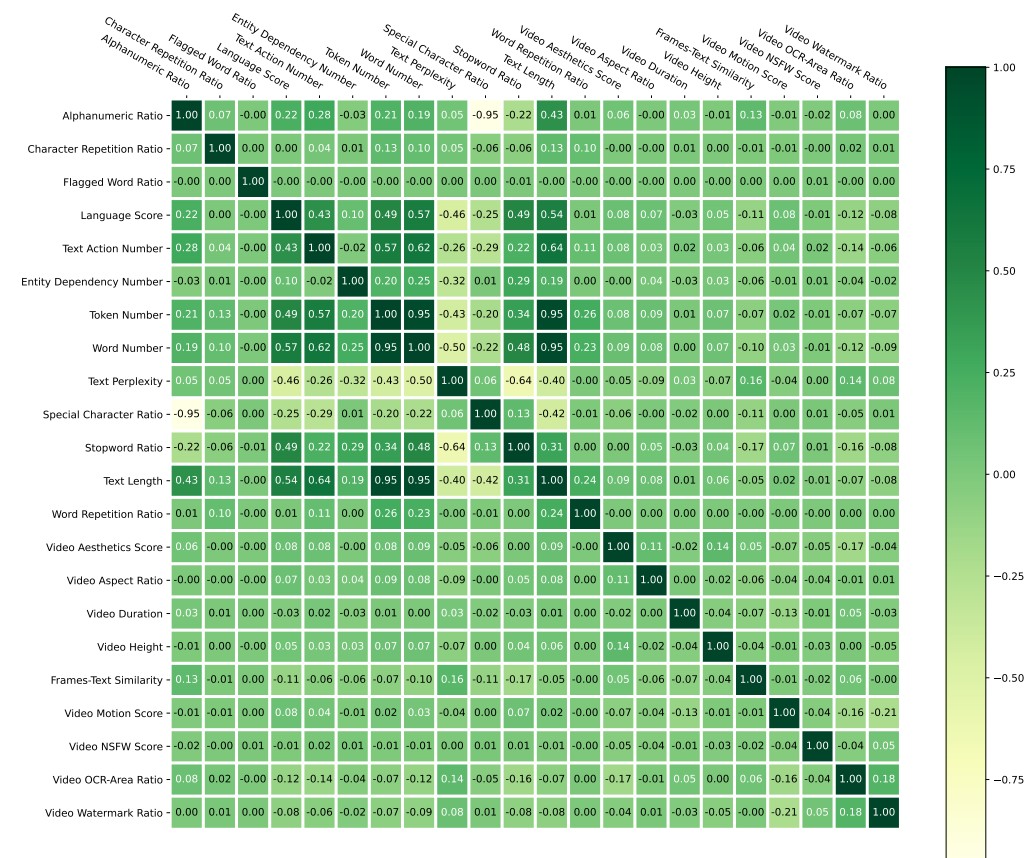

Figure 6: Pearson correlation coefficients for OP statistics in text-to-video generation.

In the second approach, we categorize the evaluation metrics based on their correlations, as illustrated in Figure 7 for image-to-text generation and Figure 8 for text-to-video generation. In image-to-text generation, several evaluation metrics are highly specific and possess unique characteristics, such as those for OCR and coding tasks, resulting in a greater number of categories upon classification. In contrast, VBench's evaluation metrics can be broadly divided into several classes, including static video metrics (e.g., subject and background consistency), dynamic metrics (e.g., dynamic degree), and video quality indicators (e.g., aesthetic quality and imaging quality).

Notably, the dynamic degree negatively correlates with many other metrics, particularly those favoring static videos, thus preventing videos without movement from being rated as optimal. Based on these observations, for text-to-video generation, we apply a hierarchical clustering algorithm (Ward Jr, 1963) to classify the VBench metrics into three categories based on their correlations: static video metrics, dynamic video metrics, and video quality along with video-text matching. We then select the best-performing OP for each of these three metric categories, where each excels in different aspects. These selected OPs are subsequently combined and experimentally evaluated together.

## C.4 Recipes Based on Correlation Analysis

In addition to selecting the overall best-performing OPs, we aim to identify operators with distinct advantages to explore whether combining these operators can synergistically leverage their strengths for improved outcomes. We selected operators with unique strengths based on **correlation information** obtained from single-operator experiments. Detailed correlation analyses can be found in Appendix C.3.

Specifically, for the image-to-text generation, we categorize the OPs by calculating correlations between their statistics. We then select representative operators within each category: `TextActionFilter` for text, `ImageNSFWFilter` for images, and

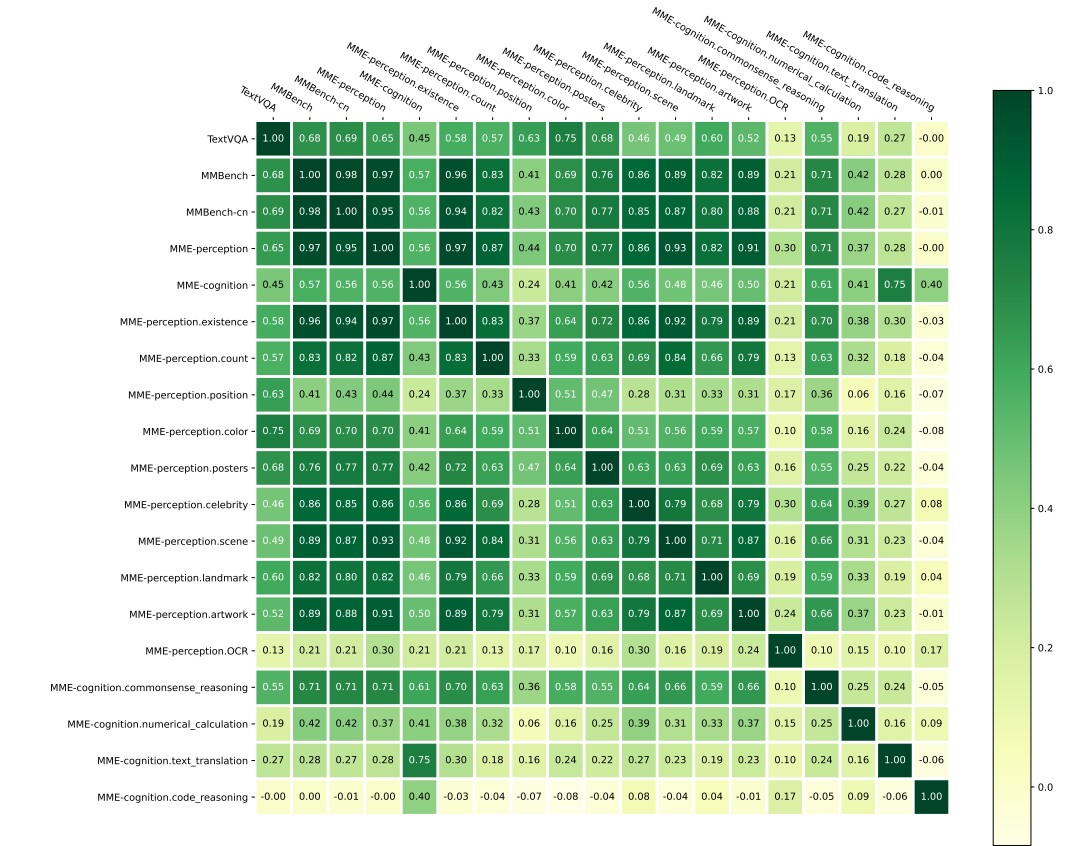

Figure 7: Pearson correlation coefficients for each dimension in TextVQA, MMBench, and MME.

`PhraseGroundingRecallFilter` for combined text-image relevance. For the text-to-video generation, based on single-operator experiments, we categorize the metrics of VBench (Huang et al., 2024) into three classes according to their relevance and select the best-performing operator from each category. The chosen operators are `VideoMotionScoreFilter`, which excels in video static features; `VideoDurationFilter`, which is superior in dynamic features; and `VideoAestheticsFilter`, which exhibits the best composite performance in video quality and video-text matching. We summarize the experiment results in Figures 9(a) and 9(b).

> **Observation 8 (Effect of Orthogonal Combination)**
>
> Combining OPs that excel in orthogonal dimensions on model or data does not guarantee complementary effects; rather, it is more likely that they will impede each other's performance.

As depicted in Figures 9(a) and 9(b), regardless of how these top-performing OPs are combined, they ultimately reduce the model's performance in both image-to-text and text-to-video generation. This observation challenges the naive assumption widely used in existing SOTA works, that various intuitively useful data cleansing actions, when stacked serially, can synergistically enhance performance.

## C.5 SETTING AND ABLATION STUDY BASED ON T2V-TURBO

The results shown in Table 3 represent the enhancements we achieved based on T2V-Turbo (Li et al., 2024a). T2V-Turbo applies LoRA (Hu et al., 2021) to VideoCrafter-2.0 (Chen et al., 2024b) and is trained on the WebVid (Bain et al., 2021) dataset, using VideoCrafter-2.0 as a teacher for distillation and incorporating reinforcement learning with rewards for the generated videos. The loss $L$ of T2V-Turbo is defined as:

$$L = L_{CD} - 1 \times R_{img} - 2 \times R_{vid}, \qquad (3)$$

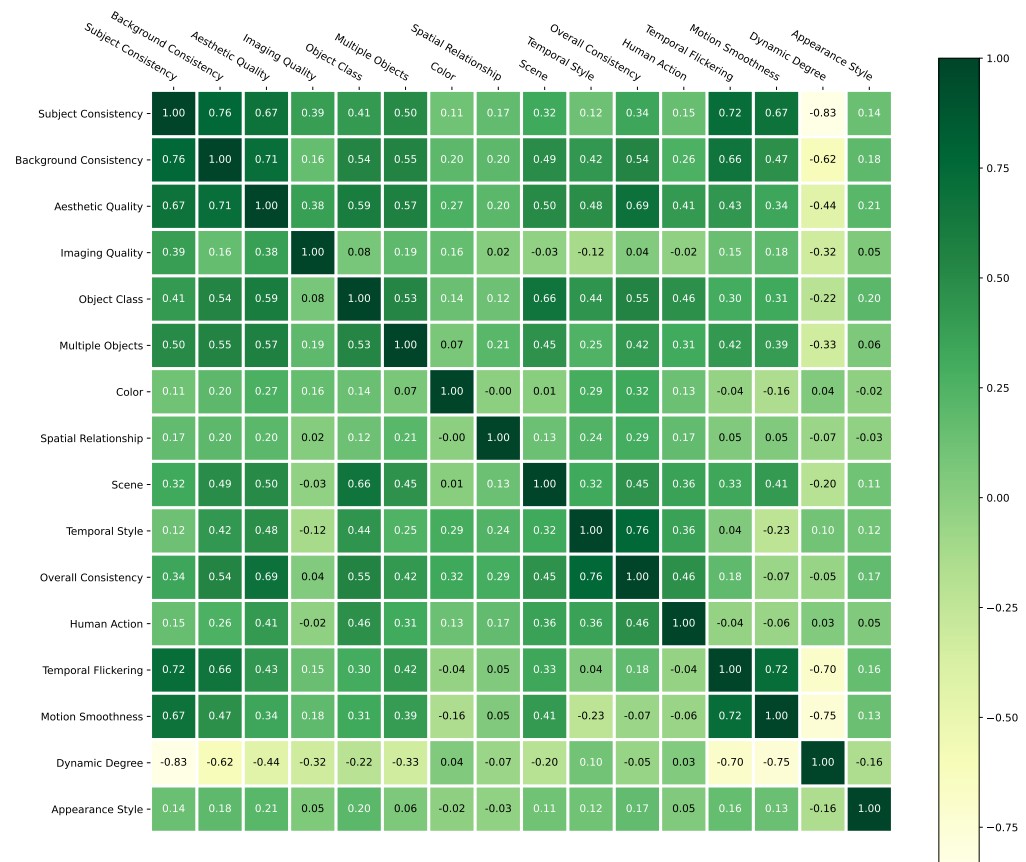

Figure 8: Pearson correlation coefficients for evaluation metrics in VBench.

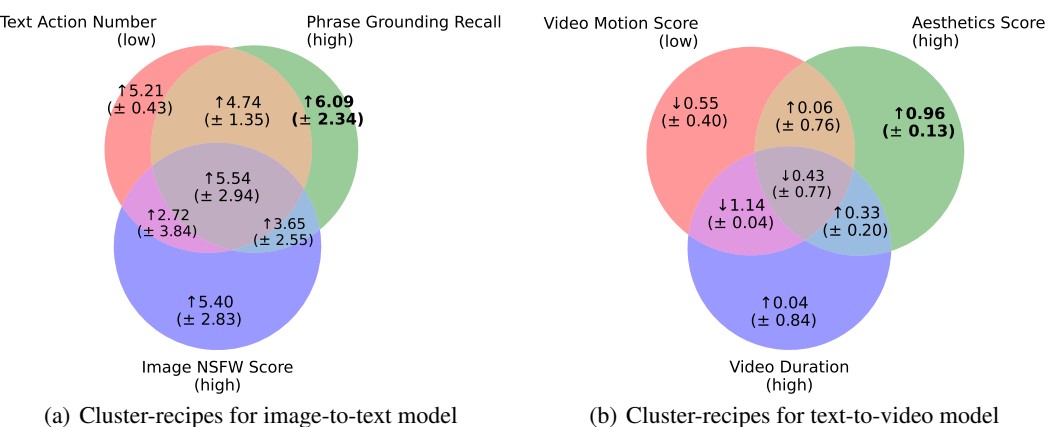

(a) Cluster-recipes for image-to-text model

(b) Cluster-recipes for text-to-video model

Figure 9: The improvements (%) of recipes with different OP combinations. The cluster recipes stem from the combinations of the best OPs in 3 categories. In the image-to-text generation, due to the limited data volume in the top-level data pool, the size of $\mathcal{P}_\mathcal{S}$ decreased from 200k to 126k in the cluster recipes.

where $L_{CD}$ is the loss of consistency distillation (Song et al., 2023), $R_{img}$ is the reward of image quality of generated video frames from HPSv2.1 (Wu et al., 2023) and $R_{vid}$ is the reward of video quality from InternVideo2 (InternVid2 S2) (Wang et al., 2024a). In the paper, we make the following cumulative modifications to T2V-Turbo:

1. **Proposed Data Pool**. Utilizing the optimal data pool identified in Section 4.4, which consists of 147k instances with low video NSFW scores and high frame-text similarities, we replace the WebVid data and train for 6 epochs based on the insight from Section 4.4.

2. **Initialization for LoRA**. We use the T2V-Turbo model to initialize the parameters for training with LoRA.

3. **Self-Distillation**. When we initialize with the T2V-Turbo model, the student model is already outperforming the teacher model, VideoCrafter-2.0, potentially leading to unstable training. To mitigate this, we use the T2V-Turbo model itself as the teacher to ensure that the reinforcement learning does not diverge excessively.

4. **Real-Data Loss**. To enhance the role of the proposed data pool during training, we add a real-data loss between the generated videos and the input videos to the distillation loss and reward loss. Furthermore, we set the weights of both the real-data loss and the distillation loss to 0.5. The modified loss from Equation 3 can be specified as:

$$L = 0.5 \times L_{CD} + 0.5 \times L_{real} - 1 \times R_{img} - 2 \times R_{vid}, \tag{4}$$

where $\times L_{real}$ is the real-data loss. Training on 147k instances with 6 epochs, we obtain the model, *Data-Juicer (T2V, 147k)*.

5. **Self-Evolution**. To validate the scalability of our methodologies, we expanded the original data pool and, by applying the `video_aesthetics_filter` and `video_motion_score_filter` to the existing recipes, we select an additional 228k instances to evolutionary self-distill *Data-Juicer (T2V, 147k)*. Meanwhile, in order to further amplify the impact of data, we have reduced the emphasis on reinforcement learning, thereby:

$$L = 0.5 \times L_{CD} + 0.5 \times L_{real} - 0.2 \times R_{img} - 0.4 \times R_{vid}. \tag{5}$$

We finally get *Data-Juicer (DJ, 228k)* in 4 epochs training.

| Model | Total Score (%) | Quality Score (%) | Semantic Score (%) |
|---|---|---|---|
| T2V-Turbo (VC2) | 81.01 | 82.57 | 74.76 |
| + Enhanced Data Pool | 81.84 | 83.40 | 75.60 |
| + Initialization for LoRA | 81.82 | **83.47** | 75.19 |
| + Self-Distillation | 79.16 | 79.48 | 77.92 |
| + Real-Data Loss | 82.10 | 83.14 | 77.93 |
| + Self-Evolution | **82.53** | 83.38 | **79.13** |

Table 7: Our model undergoes ablation experiments on the VBench leaderboard evaluation. Each '+' sign in the row indicates that the modification is added on top of the previous row's configuration.

Table 7 presents the ablation experiments of our modifications in the VBench evaluation. It is clear that when we replace the WebVid data with our proposed data pool, the model experiences a notable improvement, with the total score increasing from 81.01% to 81.84%. Subsequently, initializing the LoRA training parameters with the T2V-Turbo model does not lead to further enhancements in model performance. We suspect this might be because the teacher model is less effective than the T2V-Turbo model. Therefore, we use the T2V-Turbo model for self-distillation. While this method effectively raises the semantic score, it results in unstable video generation characterized by significant temporal flickering, which severely lowers the video quality. To counteract this, we add a real-data loss with the input data to secure the quality of the generated videos. Moreover, we evolve the model by using our trained model as the teacher model for continuous training on additional data, rather than T2V-Turbo. Ultimately, as we continue to enhance both the quality and semantic scores, we establish a new state-of-the-art.

### C.6 FULL RESULTS ON VBENCH LEADERBOARD

Backed by our proposed methodology and experiments, our Data-Juicer (DJ, with 228k) model refreshes the state-of-the-art on VBench (Huang et al., 2024), surpassing models like Gen-3 (RunwayML, 2024) and Kling (Kuaishou, 2024), as illustrated in Figure 10 and detailed in Table 3.

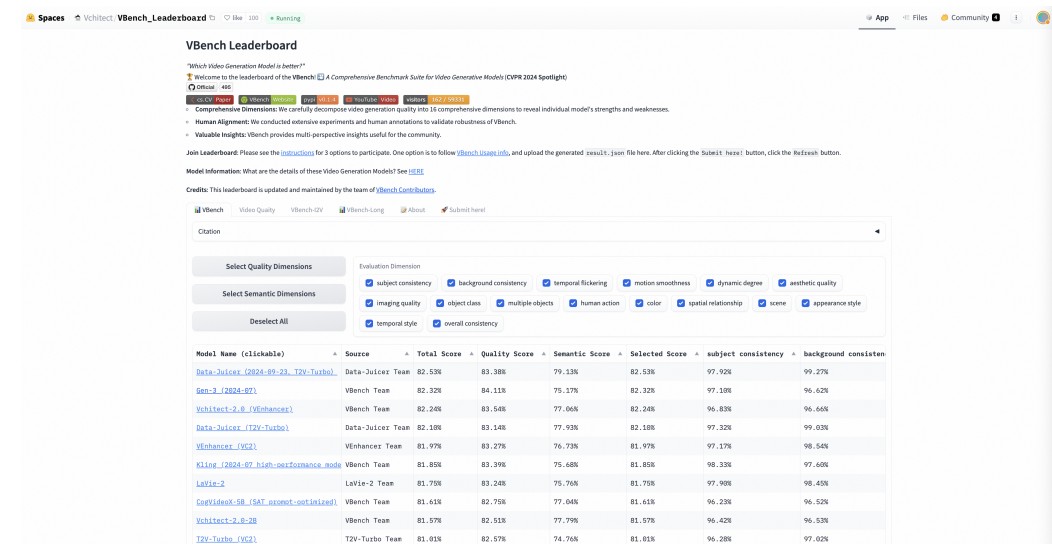

Figure 10: The VBench Leaderboard as of September 23, 2024, illustrating the rank 1 achievement empowered by our data-model co-development workflow.

| Models (Ranked by leaderboard) | Board Avg. (%) | Uniform Avg. (%) | Quality Avg. (%) | Semantic Avg. (%) |
|---|---|---|---|---|
| 1. **Data-Juicer (DJ, 228k)** | **82.53** | **81.26** | 83.38 | **79.13** |
| **Data-Juicer (T2V, 147k)** | 82.10 | 80.54 | 83.14 | 77.93 |
| 2. Gen-3 (RunwayML, 2024) | 82.32 | 79.64 | **84.11** | 75.17 |
| 3. VEnhancer (VC2) (He et al., 2024a) | 81.97 | 80.00 | 83.27 | 76.73 |
| 4. Kling (2024-07) (Kuaishou, 2024) | 81.85 | 79.54 | 83.39 | 75.68 |
| 5. LaVie-2 (Wang et al., 2023a) | 81.75 | 79.50 | 83.24 | 75.76 |
| 6. CogVideoX-5B (Yang et al., 2024) | 81.61 | 79.90 | 82.75 | 77.04 |
| 7. Vchitect 2.0-2B (Vchitect, 2024) | 81.57 | 80.15 | 82.51 | 77.79 |
| 8. T2V-Turbo (VC2) (Li et al., 2024a) | 81.01 | 78.67 | 82.57 | 74.76 |

Table 8: Leading models on the VBench leaderboard as of Sep 23, 2024. "Board Avg." denotes the weighted average of normalized scores across 16 metrics defined by VBench. "Quality Avg." represents the aggregated scores from 7 video quality metrics, whereas "Semantic Avg" aggregates scores from 9 metrics evaluating the consistency between prompts and generated videos. "Uniform Avg." indicates the simple average of scores related to quality and semantic metrics. *Data-Juicer (T2V, 147k)* is based on T2V-Turbo distillation training on 147k instances. In order to demonstrate the scalability of our methodologies, we then further self-distilled our model to obtain *Data-Juicer (DJ, 228k)* on other 228k instances.

## C.7 EXPERIMENTS ON CLIP MODELS

To further assess the robustness of our sandbox, we extended our experiments to the image-text similarity and classification tasks using the CLIP model (Radford et al., 2021). We utilize data from the small track of the DataComp competition (Gadre et al., 2023) and adhere to its evaluation metrics, which include 40 distinct evaluation subsets. Due to some broken links, we successfully downloaded 85.2% of the dataset, resulting in a total of 10.9 million samples as our $\mathcal{D}$. All baseline models were trained on an equivalent volume of data as used in the contrastive experiments, sampled randomly from this dataset. As illustrated in Figure 2, the experiments are also conducted in three phases: first, single-operation processing; second, multi-operation processing; and finally, scaling experiments with **increased data volume across various model sizes and computational scales**.

| OP-Generated Statistics | Average Performance Changes (%) | | |
|---|---|---|---|
| | Data Pool (Low) | Data Pool (Mid) | Data Pool (High) |
| *CLIP Image-Text Similarity* | -32.57 | -6.39 | **39.53** |
| *BLIP Image-Text Similarity* | -24.28 | 1.82 | **25.39** |
| *Image NSFW Score* | **12.18** | 1.28 | -18.38 |
| *Word Number* | -18.65 | 0.74 | **9.78** |
| *Stopword Ratio* | -4.28 | -3.33 | **8.97** |
| *Special Character Ratio* | **8.86** | 4.15 | -16.03 |
| *Phrase Grounding Recall* | **7.79** | 1.85 | -10.60 |
| *Text Length* | -8.31 | 1.81 | **7.29** |
| *Character Repetition Ratio* | 1.99 | 0.04 | **6.63** |
| *Image Aspect Ratio* | 4.93 | -4.55 | **5.87** |
| *Text Perplexity* | **5.27** | 2.46 | -9.56 |
| *Image Width* | -6.66 | 4.97 | **5.23** |
| *Image Height* | -4.03 | **5.02** | 0.89 |
| *Image Size* | -12.11 | **5.00** | 2.87 |
| *Image Aesthetics Score* | -9.61 | -8.13 | **4.64** |
| *Image Watermark Score* | **3.84** | -3.74 | -4.72 |
| *Flagged Word Ratio* | **3.66** | 3.47 | 1.59 |
| *Entity Dependency Number* | -5.53 | -0.39 | **2.50** |
| *Word Repetition Ratio* | -3.16 | -0.91 | **1.84** |
| *Alphanumeric Ratio* | -2.55 | **1.65** | 0.63 |
| *Token Number* | -6.35 | **1.44** | 0.27 |

Table 9: The complete OP ranking, including their statistical dimensions and the improvements relative to the baseline for image-text similarity/classification task. We consider three splits with low, middle, and high statistical values for each OP. The baseline used is based on random sampling with equal data volume.

Table 9 presents the results from single-operation processing. From which we can find that the top three performing operations are `image_text_similarity_filter`, `image_text_matching_filter`, and `image_nsfw_filter`.

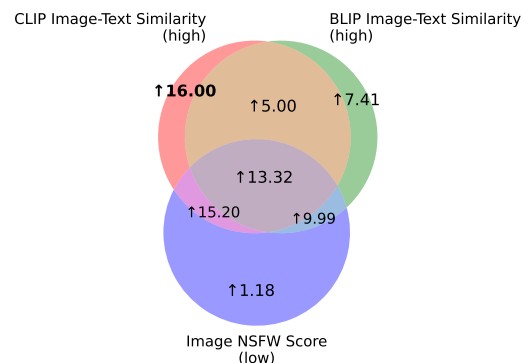

Figure 11: Percentage improvements of recipes with different operation combinations. The top-performing recipes originate from combinations of the best three operations. Due to data limitations in the top-level data pool, the training dataset size is reduced to 880k for the top recipes. Correspondingly, the default computation scale of DataComp is adjusted to 0.25 expansion rate for the sandbox experiment.

In the multi-operation processing phase, we evaluated all possible combinations of the top three operations. The results, illustrated in Figure 11, show that using the `image_text_similarity_filter` operation alone outperforms other combinations. This observation may be attributed to the filtering process relying on a high-quality CLIP model to

train another CLIP model, effectively creating a form of model distillation. Consequently, the `image_text_similarity_filter` operation consistently dominated the performance outcomes of the other operations.

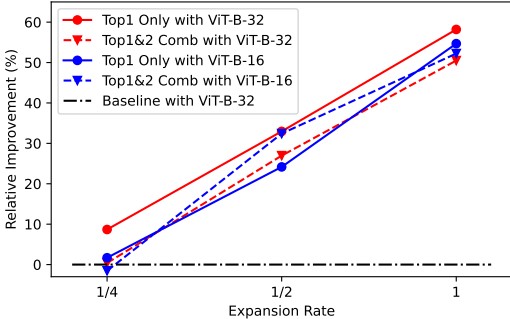

Figure 12: Relative improvement over the baseline with different FLOPS and computation scaling. "Top1 Only" refers to applying the recipe using only the `image_text_similarity_filter`. "Top1&2 Comb" indicates the use of both `image_text_similarity_filter` and `image_text_matching_filter`. "ViT-B-32" and "ViT-B-16" are image encoders of different version of CLIP, with FLOPS of 8.82G and 35.13G respectively.

Finally, we validated whether our findings in the small-scale sandbox experiments could generalize to larger scales from a scaling law perspective. Specifically, we increased the data volume from 880k to 2,683k and examined our conclusions across various models (from ViT-B-32 to ViT-B-16) and different computational scales ($1/4$, $1/2$ and $1x$ relative to the default computation of the DataComp-Small Track). As depicted in Figure 12, the `image_text_similarity_filter` operation, identified as the best performer in small-scale experiments, continued to outperform the recipe, which combines `image_text_similarity_filter` and `image_text_matching_filter`.

Furthermore, Figure 12 illustrates notably consistent improvements when both model and computational scales are increased. The linear growth in relative improvement with the exponential increase of computation aligns with known scaling laws (Cherti et al., 2023), which suggest that expanding model size and data, alongside computational scaling, yields consistent performance gains.

## C.8 OVERVIEW OF THE SANDBOX EXPERIMENTS

The table 10 provides an overview of our sandbox experiments on image-to-text generation, text-to-video generation, and image-text similarity and classification tasks.

| Sandbox Scenario | Image-to-Text Generation | Text-to-Video Generation | Text-Image Foundation Model (New in Rebuttal) |
|---|---|---|---|
| **Main Effectiveness Evidence** | Optimal recipe derived from small data pools (Sec. 4.2) achieves superior model performance in larger data pools (Sec. 4.5) | Optimal recipe from small data pools (Sec. 4.2) results in VBench-Top1 model with varying data size, model scales, and model architectures (Sec. 4.5) | Best recipe identified in the model with fewer FLOPS maintains optimal performance with increased model FLOPS and compute resources. |
| **Model Scale Range** | MGM-2B in all experiments | EasyAnimate to T2V-Turbo (Heterogeneous architectures) | CLIP: ViT-B-32 to ViT-B-16 (Different FLOPS) |
| **Data Scale Range (w.r.t Distinct Dataset Size)** | 126k to 200k | 40k to 147k and 228k | 880k to 2,683k |
| **Compute Scale Range (w.r.t Number of Trained Sample)** | 1 to 8 Epochs (Sec. 4.4) | 1 to 10 Epochs (Sec. 4.4) | 4 to 14 Epochs |

Table 10: Overview of sandbox scenarios and their effective evidence across different scales.

