# OpenReview forum: "Data-Juicer Sandbox: A Comprehensive Suite for Multimodal Data-Model Co-development"
_ICLR.cc/2025/Conference — Submitted to ICLR 2025_

### Official Review · Reviewer_AxLt · 2024-10-30

**Soundness:** 2
**Presentation:** 2
**Contribution:** 3
**Rating:** 5
**Confidence:** 4

**Summary:**

In their work, authors develop a method for co-development of multi-modal data and generative models. Data Juicer Sandbox defines a pipeline for composing data from available data pools by testing various data processing operators (OP) with regard to how those improve the quality of dataset composition for model training. This probe-analyse-refine workflow computes various stats by applying OPs to data pools and employing low cost model training to compute evals on trained models, taking the OPs leading to top performing evals for combining those OPs to create datasets at larger scales, which should further boost model training. Authors test their approach on MiniGemini (image-to-text), EasyAnimate and T2V-Turbo (text-to-video) and ModelScope, evaluating model performance on benchmarks like MMBench and VBench to provide evidence that dataset composition procedure leads to performant models. Authors conclude that observed model performance delivers evidence for the effectiveness of the prooposed data-model co-development framework.

UPDATE: score raised to 5 after rebuttal following experiments with CLIP and providing info on compute involved in video modelling experiments.

**Strengths:**

Authors address important question of how to systematically search for operations that improve dataset quality, composing datasets that result in stronger models when used for training. The workflow is open-sourced, and the study seems reproducible.

**Weaknesses:**

Central weakness is that the study attempts to make a strong claim (the presented pipeline for data-model co-development leading to dataset and model improvement) while not having proper validation via model comparison. Authors take models on a fixed reference scale and make a comparison to their method. No attempt is made to match compute or scales when doing comparison, which makes model comparison inconclusive. Authors do not attempt to derive a scaling law over model and data scales involved in their pipeline, which would at least help to understand whether claim of model improvement is consistent across at least a modest scale span. Without such experiments, evidence is weak and strong claims for general usefulness are not substantiated enough.

Important citations of previous works that attempt to design methods for systematic search of better datasets are missing. Just few examples: [DataComp, NeurIPS 2023](https://openreview.net/forum?id=dVaWCDMBof), [MetaCLIP, ICLR 2024](https://openreview.net/forum?id=5BCFlnfE1g). In DataComp for instance it is demonstrated how data and model quality should be measured across scale span.

Further weakness is paper organization. Many important experimental and method workflow design decisions and also important result details are shifted to supplementary, so that large fraction of main part reads like a coarse overview. To assess the work, reader has to go to Appendix, as main text seems to be insufficient.

**Questions:**

Why - instead of taking models that are rather not yet well established as references - do authors not take strong open reference models,  eg openCLIP, Stable Diffusion, etc, and show the strength of their approach by demonstrating how to obtain eg stronger CLIP models across various scales? That would give the method much better support, as CLIP is a strong reference trained across various open data with measurements across scales already available. That would be a clear way to show benefits if authors method would provide evidence for obtaining strong CLIP models compared to existing well established references using their data operations search procedure, deriving a scaling law for their method.

---

> ### Author Response · Authors · 2024-11-25
> **Response to Reviewer AxLt (part 1/3)**
>
> Dear Reviewer AxLt,
>
> we sincerely appreciate the time and effort you have dedicated to reviewing our paper, as well as the valuable feedback and suggestions you provide! However, after carefully reading your response, we found *there may be misunderstanding and mismatched score of our work*. In light of this, we address your comments one by one in our rebuttal and hope that you will consider re-evaluating our paper.
>
> ---
>
> ## [W1, W2 & Q1] Validation via model comparsion across scale span, and adding experiments for CLIP models and related citations
> + **`(Clarification of our model validation experiments)`** We respectfully disagree with the reviewer's opinion about "_not having proper validation via model comparison_" and "_whether claim of model improvement is consistent across at least a modest scale span_". After carefully reading all your comments,  we found you may have misunderstood the validation setting of our works. Specifically, as the table below shown, we have covered wide application examples:
>     - (1) achieving superior performance with less data in LLaVA-like MGM models,
>     - (2) taking the top spot on the VBench leaderboard with DiT-Based EasyAnimate and T2V-Turbo models, outperforming many commercial closed-source models, and
>     - (3) during the rebuttal period, adding insights for CLIP models under varying computation scales (further details are provided later).
>     - A summarization Table covered different dimensions is shown below:
>
> | **Sandbox Scenario**                                         | Image-to-Text Generation                                                                                               | Text-to-Video Generation                                                                                                                         | Text-Image Foundation Model *(New in Rebuttal)*                                                                              |
> |--------------------------------------------------------------|-------------------------------------------------------------------------------------------------------------------------|------------------------------------------------------------------------------------------------------------------------------------------------|------------------------------------------------------------------------------------------------------------------------------|
> | **Main Effectiveness Evidence**                              | Optimal recipe derived from small data pools (Sec. 4.2) achieves superior model performance in larger data pools (Sec. 4.5) | Optimal recipe from small data pools (Sec. 4.2) results in VBench-Top1 model with varying data size, model scales, and model architectures (Sec. 4.5) | Best recipe identified in the model with fewer FLOPS maintains consistent performance advantages with increased model FLOPS and compute resources. (Sec.C.7) |
> | **Model Scale Range**                                        | MGM-2B in all experiments                                                                                            | EasyAnimate to T2V-Turbo *(Heterogeneous architectures)*                                                                                         | CLIP: ViT-B-32 to ViT-B-16 *(Different FLOPS)*                                                                              |
> | **Data Scale Range** *(w.r.t Distinct Dataset Size)*             | 126k to 200k                                                                                                           | 40k to 147k and 228k                                                                                                                             | 880k to 2,683k                                                                                                                |
> | **Compute Scale Range** *(w.r.t Number of Trained Sample)*   | 1 to 8 Epochs *(Sec. 4.4)*                                                                         | 1 to 10 Epochs *(Sec. 4.4)*                                                                                                                          | 4 to 14 Epochs                                                                                                              |
>
> (To be continued.)

---

> ### Author Response · Authors · 2024-11-25
> **Response to Reviewer AxLt (part 2/3)**
>
> (Continued from the previous page, [W1, W2 & Q1])
>
> + **`(Comparision to related works)`** Thanks for your suggestion, the references you provide are highly valuable, and we have included them in the related works section of our revised paper. In short, our work differs them in goals and methdologies:
>     - DataComp introduces a benchmark to filter out high-quality data from 12.8 billion image-text pairs in Common Crawl to train better CLIP models, considering Filter Operators such as CLIP score, image size, and caption length. MetaClip aims to reproduce CLIP's data by introducing a raw data pool and finding a balanced subset based on CLIP's concepts.
>     - Unlike these data-centric approaches that isolate the model and training settings, focusing solely on the quality and scale of training datasets, the proposed sandbox suite emphasizes systematic methodologies for data-model co-development, considering both data and models equally important.
>     - Specifically, we incorporate performance signals from sandbox reference models on many downstream tasks, conducting importance and correlation analysis to link data pools and these model metrics. Additionally, we explored more model types beyond CLIP, such as LLaVA-like and DiT-based models for image-to-text and text-to-video tasks, and identified better training datasets for these models using our workflow.
>
> + **`(New experiment results for CLIP)`** In addition to the practical examples from two distinct fields already presented in our paper, we have, within the short rebuttal period, swiftly applied the sandbox workflow to the new CLIP model scenario. By reusing parts of the datacamp's training and evaluation code, we were able to adapt this quickly and further explore the sandbox's potential in compute scaling, substantiating the suite's adaptability. **More detailed Tables and Figures are provided in the revised Section C.7.** Here we summarize the experiments as follows:
>     - **(Setting):** We utilize data from the small track of the DataComp competition~\citep{datacomp} and adhere to its evaluation metrics, which include 40 distinct evaluation subsets. Due to some broken links, we successfully downloaded 85.2\% of the dataset, resulting in a total of 10.9 million samples as our $\mathcal{D}$. All baseline models were trained on an equivalent volume of data as used in the contrastive experiments, sampled randomly from this dataset. As illustrated in Figure 2, the experiments are also conducted in three phases: first, single-operation processing; second, multi-operation processing; and finally, scaling experiments with \textbf{increased data volume across various model sizes and computational scales}.
>     - **(Finding Best Recipe):** Table 9 presents the results from single-operation processing. From which we can find that the top three performing operations are _image_text_similarity_filter_, _image_text_matching_filter,_ and _image_nsfw_filter_. In the multi-operation processing phase, we evaluated all possible combinations of the top three operations. The results, illustrated in Figure 11, show that using the _image_text_similarity_filter_ operation alone outperforms other combinations. This observation may be attributed to the filtering process relying on a high-quality CLIP model to train another CLIP model, effectively creating a form of model distillation. Consequently, the _image_text_similarity_filter_ operation consistently dominated the performance outcomes of the other operations.
>     - **(Scaling Study):** Finally, we validated whether our findings in the small-scale sandbox experiments could generalize to larger scales from a scaling law perspective. Specifically, we increased the data volume from 880k to 2,683k and examined our conclusions across various models (from ViT-B32 to ViT-B-16) and different computational scales (1/4, 1/2 and 1x relative to the default computation of the DataComp-Small Track). As depicted in Figure 12, the _image_text_similarity_filter_operation_, identified as the best performer in small-scale experiments, continued to outperform the recipe, which combines _image_text_similarity_filter_ and _image_text_matching_filter_. Furthermore, Figure 12 illustrates notably **consistent improvements when both model and computational scales are increased.** **The linear growth in relative improvement with the exponential increase of computation** aligns with known scaling laws (Cherti et al., 2023), which suggest that expanding model size and data, alongside computational scaling, yields consistent performance gains.

---

> ### Author Response · Authors · 2024-11-25
> **Response to Reviewer AxLt (part 3/3)**
>
> ## [W3] Paper Organization "... important ... decisions ... result details are shifted to supplementary ..."
> Thank you for your comments and suggestions! We have taken your feedback seriously and have made several improvements to the organization in the revised version. By simplifying wording and expressions and effectively utilizing space, we have brought forward some necessary details. However, we would like to clarify why we chose to place fewer detailed settings in the main text. Fundamentally, the workflows and insights mentioned are merely instantiations of the sandbox platform's capabilities, serving as proof of concept for data-model co-development with the help of the Data-Juicer Sandbox.
>
> To maintain the focus on its conceptual utility, we intentionally deferred some details. Specifically, we clarify the dual focus of our paper as follows, integrating both the Sandbox platform and the insights derived from it. We believe these elements are inseparably intertwined, each crucial for conveying the full value of our work:
>
> + **Platform Development as a Foundation for Insights**: The platform serves as the essential groundwork for deriving experimental insights. Due to the current absence of ready-made open-source middleware infrastructure specifically designed for data-model co-development, we had to first develop the necessary codebase and platform. This step was crucial for supporting subsequent empirical explorations. Recognizing this gap and to facilitate and accelerate related research, we committed significant efforts to design the system with a strong emphasis on reusability, and we have released it open-source for the community. Without this platform, broader exploration of such topics in an open-source context would remain challenging.
> + **Insights as Demonstrations of Platform Utility:** The experimental insights presented in the paper highlight key potential use cases of the platform. They serve as critical demonstrations of the platform's utility, usability, and effectiveness, significantly aiding readers' understanding and boosting confidence in its applicability.
> + **How These Demonstrations Help Users to use the Platform:** As you mentioned in Q3, a user or organization considering such a platform would weigh cognitive (migration) costs and, more importantly, the ability to effectively conduct data-model co-development. Compared to existing related open-sourced works, our platform enables convenient, efficient, and scientifically rigorous insights extraction that can be applied across a broad range of scenarios (more discussion on related works has been added in the revision, Appendix A). For recap, the application examples have included achieving superior performance with less data in LLaVA-like MGM models, taking the top spot on the VBench leaderboard with DiT-Based EasyAnimate and T2V-Turbo models, outperforming many commercial closed-source models, and during the rebuttal period, adding insights for CLIP models under varying computation scales (further details are provided in the response to Q11). These insights and results have already attracted inquiries and collaborative interest from numerous open-source users and industry teams (specific entities cannot be disclosed due to anonymity requirements).
> + **The advantages from Open-source Community**: Additionally, the transparency and rapid evolution of the open-source community are among the suite's advantages. We continually integrate representative development libraries and embrace the evolving open-source landscape. For instance, the Data-Juicer system now includes over 100 operators covering various modalities, doubling the number from the initial Sandbox phase. Integrated libraries like modelscope-trainer and LLaVa training codes are also continuously iterating, further accelerating migration and reuse within the suite.
> + **Platform Details in the Main Text of the Paper**: Considering the diversity of the ICLR community and the relevance and timeliness of the topic, we intentionally limited the focus on platform specifics within the main paper, as general machine learning readers may not be interested in code-level design and implementation details. Instead, this information is included in the appendix along with the source code and accompanying documentation, which have been provided in the submitted anonymous link and are continuously maintained online. We also incorporate your suggestions, such as mentioned that users can `python tools/sandbox_starter.py --config configs/demo/sandbox/sandbox.yaml` to utilize easy one-line commands and simple configurations to switch between many built-in hooks and capabilities.
>
> ---
>
> ## Closing Remarks
> We would like to express our sincere gratitude for your suggestions and queries. *We hope that our responses have addressed your comments and you can consider re-evaluating our paper.* We look forward to any additional feedback you might have. Thank you again for your valuable input!

---

> ### Comment · Reviewer_AxLt · 2024-11-30
>
> Thanks for the thorough rebuttal.
>
> I appreciate the time you have invested in additional experiments with CLIP. I think the placement in Appendix is rather unfortunate, as to me it seems to be the only part of the current work that attempts to make model comparison while controlling for total amount of compute spent for training. I would thus recommend to put it more prominently.
>
> While seeing the merit of data and model co-development put forward by the work, I still struggle with model comparisons presented in the study. Eg Table 3 emphasizes Data Juicer derived model on rank 1 - how is this model total compute spent for training relates to the other models on the board? Is there a way to provide info on model scale / samples seen combination during training (or are authors assuming those are already well aligned between the models?) or give total compute spent for compared models? Best would be a plot with total estimated pretraining FLOPS on x axis that shows model performance in relation to their compute - how it is done conventionally when providing proper model performance comparison. It might be the case that better ranked Data Juicer models have correspondingly more compute put into training, which would be a confound when arguing for the effect of Data Juicer pipeline in bringing the improvement along. If nitpicking, compute spent on Data Juicer pipeline has to be taken into account itself, if it is substantial, but I would be already happy if seeing compute comparison just for pretraining.
>
> I will consider feedback to this comments when adjusting my final score.

---

> > ### Author Response · Authors · 2024-12-02
> > **Second-Round Reply to Reviewer AxLt (part 1/2)**
> >
> > Dear Reviewer AxLt,
> >
> > Thank you for your time and constructive feedback! We appreciate your recognition of our efforts in conducting additional experiments with CLIP. Based on your suggestions, we have outlined the following responses and corresponding modifications:
> >
> > ---
> >
> > 1. `Prominence of CLIP Experiments`
> >
> > We agree that the CLIP experiment results should be more prominent. We will move these new results from the appendix into the main paper to emphasize their importance in the final version. Although we cannot update the revision at this stage, we briefly describe how we achieve this: merging Figure 11 into Figure 3 and Figure 12 into Figure 4, while condensing some repetitive content from other tasks (image-to-text and text-to-video) to create space. We'll also introduce a new Section 4.6 entitled "Experiments on CLIP Models" on the main page to highlight these findings.
> >
> >
> > 2. `Baseline Comparison from Compute Cost View`
> >
> > Thanks for your suggestion! We acknowledge the importance of detailing compute resources and appreciate your insights. Here’s how we address this:
> >
> >  - [**Previous Details about Compute Cost**] We have included details about the training cost in terms of dataset size and GPU times in Table 2 and Appendix B.6 regarding the Data Juicer pipeline and the training of MGM and Data Juicer T2V models, which may not have been sufficiently noticeable. Specifically, in Table 2, our MGM model shows a +2.12% average performance improvement over baseline, using approximately 51.8% of the FLOPS. In Appendix B.6, we mentioned that our experiment on a single data pool is designed to be completed within one day using a GPU. Consequently, with 8 A100 GPUs, we can run all our experiments within approximately 3 days, 7 days, and 10 days for the CLIP, image-to-text, and text-to-video experiments, respectively.  The training sample size is in the order of $O(1)$ million video-text pairs and $O(1)$ million image-text pairs in our experiments. As a reference, the pre-training cost for a cutting-edge text-to-video model like Meta's MovieGen involves 6,144 H100 GPUs, $O(100)$ million video-text pairs and $O(1)$ billion image-text pairs.
> >  - [**The Flexibility and Potential of the Pipeline to Reduce Amortized Cost**] More importantly, the proposed method enables flexibily to adjust the experiment plan based on avaliable resources, and it holds good potential to reduce the overall cost.
> >  	- Let the model training time with a full-size dataset be denoted as $T_{full}$, and we need to iterate the training $M$ times to get a preferable performance. Denote the model training time as $T_{pool}$ on a small-size sampled dataset pool with a decreased ratio of $r$ compared to the training steps of the case of full-size dataset, and let the total number of planned small-pool experiments be $m$.
> >  	- Then, the total time without our pipeline is $M \times T_{full}$, and the total time with our pipeline is $(T_{full} + m \times T_{pool}) \approx (1+mr)\times T_{full}$. We can see that it is practical to achieve $(1+mr) \le M$, where $M$ usually greater than 2 because model developers typically iterate on multiple model and dataset versions for comparison. The factor $m$ positively corresponds to the number of data processing operators of interest (usually in the scale of dozens), and $r$ can be much smaller than 0.01 since there is no need to make small-pool experiments to "converge", as long as we can observe enlightening changes—positive or negative—in models after targeted data intervention versus random data sampling. Besides, we can early-stop some unpromising experiment trials to speed up this process.
> >   	- *In summary, our goal is to find insights from the $m$ cost-controled experiment trials and apply them to larger-scale scenarios in a more quantitative and scientific way compared to directly develop models through $M$ costful (and usually heuristic) experiment trials.*
> >
> > (To be continued.)

---

> > > ### Author Response · Authors · 2024-12-02
> > > **Second-Round Reply to Reviewer AxLt (part 2/2)**
> > >
> > > (Continued from previous page, point 2, "Baseline Comparison from Compute Cost View".)
> > >
> > >
> > >  - [**New Details about FLOPS**]  To incorporate your suggestion and make these information more clear, we try our best to complement the pretraining FLOPS of these models and will put the new results into our final version.
> > >  	- Noting that estimating the FLOPS for some models in Table 3 and Table 8 poses several challenges due to the lack of publicly available information. For example, some of their training codes, training datasets have not been open-sourced, and specific details such as the resolution and frame rate of training videos are unavailable. Thus, we report the estimated FLOPS for the models with sufficient details about the training samples and open-sourced models below. We utilized the inference code of their released open-source models and measured the FLOPS during inference using the [calflops](https://github.com/MrYxJ/calculate-flops.pytorch) tool. We then estimated the training FLOPS by applying a factor of $\alpha$ and taking into account the disclosed quantity of training data. From which we can see that our method achieves strong performance while utilizing fewer FLOPS, especially in comparison to the third-place model, VEnhancer, which also originates from VC2, the same upstream model as ours. The second-place Gen-3 and the fourth-place Kling are omitted as they are proprietary models within commercial products.
> > > | Models            | Board Performance | Dataset Size | Inference FLOPS per Sample     | (Estimated) Number of Training Samples | (Estimated) Training FLOPS       |
> > > |-------------------|-------------|--------------|--------------------------------|--------------------------------------|--------------------------------|
> > > | Ours (stems from VC2)        | 82.53       | 228k         | 12T/sample, 320p * 8fps        | 640k                                 | $7.68 \alpha$ EFLOPS           |
> > > | Ours (stems from VC2)        | 82.10       | 147k         | 12T/sample, 320p * 8fps        | 640k                                 | $7.68 \alpha$ EFLOPS           |
> > > | VEnhancer (stems from VC2)   | 81.97       | 350k         | 478T/sample, 720p * 24fps      |  $\geq$ 350k                                 | $\geq 167.3 \alpha$ EFLOPS   |
> > > | CogVideoX-5B      | 81.61       | 1,051M       | 338T/sample, 720p * 8/16fps    | 1,051M                               | $355238 \alpha$ EFLOPS        |
> > > | Vchitect 2.0-2B   | 81.57       | -            | 252T/sample, 768p * 8fps       | -                                    | -                              |
> > >
> > >
> > > ---
> > >
> > > We are committed to enhancing transparency and clarity to strengthen our study's contributions. Your feedback is invaluable in refining our work, and we thank you for considering our responses in your final evaluation of the paper.
> > >
> > > Best, authors

---

> > > > ### Comment · Reviewer_AxLt · 2024-12-03
> > > >
> > > > I would like to thank authors for the thorough rebuttal. Compute calculations make the model comparison comprehensible, and it seems to me that at least claims of better video model quality achieved when using DataJuicer pipeline are justified, For CLIP experiments, I think same presentation of compute used to obtain the models will be useful to have a way to compare obtained CLIP models to the references (also obtained in DataComp - although compute there is fixed for each track by fixing data scale and model scale, FLOPs numbers would be very informative and allow comparison to other existing CLIP derived models)
> > > >
> > > > Given the improvements achieved during rebuttal, I would like to increase the score to 5 - to argue for strong claims in the work, one would have to present proper scaling law derived for tested models, which is not the case for the study. However, it seems that some reference comparisons provide preliminary evidence that framework is indeed useful, so it is a borderline. I am aware that this is a burden for chairs to decide, with other scores being tight.

---

> > > > > ### Author Response · Authors · 2024-12-04
> > > > > **Third-Round Reply to Reviewer AxLt (part 1/2)**
> > > > >
> > > > > Dear Reviewer AxLt,
> > > > >
> > > > > Thank you for your thoughtful and constructive feedback on our submission! We appreciate your acknowledgment of the improvements made during the rebuttal process and are pleased that you found our compute calculations helpful in clarifying model comparisons. Below we give some more clarifications about your latest response:
> > > > >
> > > > > ---
> > > > >
> > > > > 1. `CLIP FLOPs`
> > > > >
> > > > > We understand your suggestion regarding the presentation of compute metrics, particularly FLOPs, for the CLIP experiments. As you noted, such information would allow for more precise comparisons with existing models and clarify the computational resources utilized in our study, which we will incorporate into our final version. Specifically, refer to Figure 12; the FLOPs with full expansion rate are 189 $\alpha$ PFLOPS (14.78G inference FLOPS per sample) and 526 $\alpha$ PFLOPS (41.09G inference FLOPS per sample) for B-32 and B-16, respectively.
> > > > >
> > > > >
> > > > > 2. `Scaling Law Derivations`
> > > > >
> > > > > We agree that providing a scaling law would better support our claim about data-model co-development. In the experiments with CLIP, we have provided results for different computational budgets and various model FLOPs as exploration per your suggestions. However, we respectfully argue that this law derivation is beyond the current version titled _**Data-Juicer Sandbox: A Comprehensive Suite for Multimodal Data-Model Co-development**_, and it deserves to be addressed more fully in future work, for the following reasons:
> > > > >
> > > > >
> > > > > 2.1 **Our experiments focus on data-model co-development**
> > > > >
> > > > > As listed in our previous [response](https://openreview.net/forum?id=U1o9KaRgYQ&noteId=bOy8jLIIvD), we introduced a proportion factor $r$ indicating pools sampled from the full dataset $T_{full}$ for our experiments, identified the best-performing recipe, and then applied this recipe to the training of $T_{full}$ with different model architectures and models with larger FLOPS, *achieving consistent performance improvements considering data, model, and compute*. From this view, we believe the paper remains firmly within the scope of data-model co-development, rendering the claim not overstated.
> > > > >
> > > > >
> > > > > 2.2 **Scaling laws deserve a full dedicated paper**
> > > > >
> > > > > We acknowledge that there are still unresolved issues, such as determining the smallest value of $r$ that can still provide positive and stable feedback while minimizing the overhead of our pipeline. Research on scaling laws would be an excellent approach to address this problem.
> > > > >
> > > > > However, for a relatively novel topic like this, we believe that there are numerous foundational issues that need to be addressed, not only exploring the important question regarding scaling laws (especially those that can be formalized or validated on a large scale). Some examples include the development of *open-source, highly accessible software platforms* and *practical use cases* for the data-model co-development, which we have already invested considerable effort in bridging these gaps. Based on our current version, deriving formal scaling laws or discovering laws from large-scale and cross-scale scenarios (which are usually computationally expensive) may be infeasible to be fully covered in a (few) brief section(s), and we think it deserves another fully dedicated paper.
> > > > >
> > > > >
> > > > > 2.3 **The content of this work is already quite comprehensive and sufficient.**
> > > > >
> > > > > As clarified above, we have considered several timely and popular foundation models that emerged in the past year, including LLaVA-like image-to-text models and Dit-Based text-to-video models. After the rebuttal, we also incorporated the foundational pre-trained CLIP model you recommended. As a result, the total number of pages in our paper has now reached more than 30. Given the constraints of space in our current manuscript, we believe it is more appropriate to explore the scaling laws as future work.
> > > > >
> > > > > (To be continued.)

---

> > > > > ### Author Response · Authors · 2024-12-04
> > > > > **Third-Round Reply to Reviewer AxLt (part 2/2)**
> > > > >
> > > > > 3. `New theoretical explanations instead of scaling law`
> > > > >
> > > > > Instead of using a scaling law, we now provide a new justification through the following theoretical derivations, which closely demonstrate the rationality of our current experiments. Let $X_{full}$ be the improvement of a certain operator in training a model after filtering data on the full dataset $T_{full}$ compared to random selection, and let $X_{sub}$ be the improvement of the same operator in training a smaller model after filtering data on a subset of $T_{full}$ with sample ratio $r$. Then, the probability that the error introduced by our experiments on the smaller dataset is greater than $\epsilon$ can be denoted as:
> > > > >
> > > > > $$\mathbb{P}[X_{sub} - \mathbb{E}[X_{full}] \geq \epsilon]$$
> > > > >
> > > > > It is worth noting that our sandbox aims to obtain effective feedback with minimal expenditure. This feedback is often less apparent than what is obtained from the full dataset and larger model, meaning that $\mathbb{E}[|X_{full}|] \geq \mathbb{E}[|X_{sub}|]$. For operators that have a positive effect, we have $\mathbb{E}[X_{full}] \geq \mathbb{E}[X_{sub}]$. In this case, we can conclude that:
> > > > >
> > > > > $$
> > > > > \mathbb{P}[X_{sub} - \mathbb{E}[X_{full}] \geq \epsilon] \leq \mathbb{P}[X_{sub} - \mathbb{E}[X_{sub}] \geq \epsilon] \leq e^{-2\epsilon ^2 / (b-a)^2}
> > > > > $$
> > > > >
> > > > > Here we use Hoeffding's inequality with the assumption that $X_{sub} \in [a, b]$. As the sample rate $r$ increases, the variance of the improvement across different training trials decreases, which is positively related to $(b-a)^2$, and thus the right term decreases. In conclusion, the probability of the error exceeding $\epsilon$ decreases exponentially as $r$ increases. We will add this theoretical discussion to our final version to enhance the validity of our experiments.
> > > > >
> > > > > ---
> > > > >
> > > > > We are committed to enhancing transparency and clarity to strengthen our study's contributions.  Your feedback has been very helpful in enhancing the work.
> > > > >
> > > > > We believe data-model co-development is a direction worthy of diverse and further exploration. As a pioneering work to provide such a data-model co-development tool suite and workflow, we hope that the reviewer can consider our merits and increase the score if our new response addresses your remaining comments. We sincerely thank the reviewer again!
> > > > >
> > > > > Best, authors

---

### Official Review · Reviewer_Wtwc · 2024-11-02

**Soundness:** 2
**Presentation:** 2
**Contribution:** 3
**Rating:** 6
**Confidence:** 4

**Summary:**

This paper introduces a sandbox tool that allows rapid testing and analysis of multiple datasets in the training of machine learning models. It contains multiple classes that provide filtering operations and other evaluative capabilities. The paper also proposes a workflow named "Probe-Analyze-Refine". It first creates equal sized data subsets by slicing data on percentile ranks based off various data-related metrics. It then trains and evaluates models on external benchmarks to identify the best data splits, defined as the best performance on the benchmarks. It also attempts to combine the filtering operations in an attempt to create better training subsets. The paper then performs a study and presents various observations of their experiments.

**Strengths:**

The paper introduces an open-sourced tool that may be useful to the community. It demonstrates its utility by performing some data experiments; by training MGM-2B models on different slices of the original data, the authors draw conclusions about the effectiveness of various data slices. While the takeaways are not surprising, they help the community by being experiment validations of some long-held “common-knowledge”.

**Weaknesses:**

The paper needs to decide if it is trying to introduce the Data Juicer Sandbox platform or to derive insights from data tests. It starts off introducing the Sandbox and, as a reader, I expected to understand the various functionalities and automation it offers. However, it doesn’t go into details about its implementation (e.g., perhaps some code optimization of the data process), demonstrate how it helps the users (e.g., maybe the user has to write only 4 lines of code instead of 50), or even the terminology (e.g., workflow, behavior, and capabilities from Lines 125—126 are not defined and the relationship between them is unclear). It then abruptly switches to introducing how to do data splits and training models on them to test their impact on the final performance of the models. The rest of the paper then elaborates on different ways to combine filters in attempts to obtain a final data subset that will result in more efficient training and a better final model. Without understanding the focus of the paper, it is difficult to judge its overall contribution.

Overall, the paper can also benefit from simpler and cleaner presentation; words like “bespoke” (L125) can be eliminated without changing the meaning of the sentences. Some of the figures can benefit from subfigures and subcaptions. The grammar can improve but this is relatively minor.

**Questions:**

1.	L91-92: Is there a source for “traditional data-centric or model-centric strategies … resource misallocation”?
2.	Does the Data-Juicer sandbox require specific infrastructure/machines or does it work out-of-the-box in any computing environment or hardware (e.g., with TPUs)?
3.	Many companies/research labs have their own workflow for scaling data processing and training. They also study scaling laws and various data filters. What are the advantages of using Data-Juicer sandbox compared to their customized process? How can you improve to coax them to join your open-sourced framework/project?
4.	Are the "data processing operators (OPs)" defined in the article (e.g., on L186) simply data filters?
5.	Is there a reason why the data is sorted into three equal sized pools (L210 – L211) instead of just 2? Why not have 4 or 5 buckets? It would suffice to explain your rationale for your choice.
6.	How is “size of the data pool” defined? What happens if the filter results in less than required amount of data (e.g., “size” set to 200k but the result of filtering is 100k)? Please explain the terms clearer.
7. Is there a fixed computing budget for training a model on each data slice? If so, how is it defined (e.g., time budget or number of TFLOPS)?
8.	How is the “expansion rate” (L416) defined? If it means duplicating the data slice, please just mention it as such. Duplicating the data slice essentially means that the model has been trained for more epochs. Is it then fair to compare 2 models when the number of epochs trained are different?
9. How do you ensure that the models are "done" training?
10.	Related to the above question, it may be good to illustrate the training convergence of some of models on different data slices.
11. Overall, it would be good to consider the interaction between data, models, and compute instead of only the former 2.

---

> ### Author Response · Authors · 2024-11-25
> **Response to Reviewer Wtwc (part 1/3)**
>
> Dear Reviewer Wtwc,
>
> we sincerely thank you for your time and valuable feedback! Following your constructive suggestions, we conduct several improvements requiring considerable time and effort. Below, we provide detailed responses to each of your comments, with the hope that this will encourage you to lean toward the acceptance of our paper.
>
> ---
>
> ## [W1 & Q3] Focus of the Paper: Platform vs. Insight
> > "... Sandbox platform or to derive insights ... Without understanding the focus of the paper, it is difficult to judge its overall contribution." "What are the advantages of using Data-Juicer sandbox ..."
> >
>
> Thank you for your questions. We appreciate the opportunity to clarify the dual focus of our paper, which integrates both the Sandbox platform and the insights derived from it. We believe these elements are inseparably intertwined, and each is crucial for conveying the full value of our work:
>
> + **Platform Development as a Foundation for Insights**: The platform serves as the essential groundwork for deriving experimental insights. Due to the current absence of ready-made open-source middleware infrastructure specifically designed for data-model co-development, we had to first develop the necessary codebase and platform. This step was crucial for supporting subsequent empirical explorations. Recognizing this gap and to facilitate and accelerate related research, we committed significant efforts to design the system with a strong emphasis on reusability, and we have released it open-source for the community. Without this platform, broader exploration of such topics in an open-source context would remain challenging.
> + **Insights as Demonstrations of Platform Utility:** The experimental insights presented in the paper highlight key potential use cases of the platform. They serve as critical demonstrations of the platform's utility, usability, and effectiveness, significantly aiding readers' understanding and boosting confidence in its applicability.
> + **How These Demonstrations Help Users to use the Platform:** Regarding the raised Q3, a user or organization considering such a platform would weigh cognitive and migration costs and, more importantly, the ability to effectively conduct data-model co-development. Compared to existing related open-sourced works, our platform enables convenient, efficient, and scientifically rigorous insights extraction that can be applied across a broad range of scenarios (more discussion on related works has been added in the revision, Appendix A). For recap, the application examples have included achieving superior performance with less data in LLaVA-like MGM models, taking the top spot on the VBench leaderboard with DiT-Based EasyAnimate and T2V-Turbo models, outperforming many commercial closed-source models, and during the rebuttal period, adding insights for CLIP models under varying computation scales (further details are provided in the response to Q11). These insights and results have already attracted inquiries and collaborative interest from numerous open-source users and industry teams (specific entities cannot be disclosed due to anonymity requirements).
> + **The advantages from Open-source Community**: Additionally, the transparency and rapid evolution of the open-source community are among the suite's advantages. We continually integrate representative development libraries and embrace the evolving open-source landscape. For instance, the Data-Juicer system now includes over 100 operators covering various modalities, doubling the number from the initial Sandbox phase. Integrated libraries like modelscope-trainer and LLaVa training codes are also continuously iterating, further accelerating migration and reuse within the suite.
> + **Platform Details in the Main Text of the Paper**: Considering the diversity of the ICLR community and the relevance and timeliness of the topic, we intentionally limited the focus on platform specifics within the main paper, as general machine learning readers may not be interested in code-level design and implementation details. Instead, this information is included in the appendix along with the source code and accompanying documentation, which have been provided in the submitted anonymous link and are continuously maintained online. We also incorporate your suggestions, such as mentioned that users can `python tools/sandbox_starter.py --config configs/demo/sandbox/sandbox.yaml` to utilize easy one-line commands and simple configurations to switch between many built-in hooks and capabilities.

---

> ### Author Response · Authors · 2024-11-25
> **Response to Reviewer Wtwc (part 2/3)**
>
> ## [W2, Q1-2, Q4-Q10] Presentation Suggestions and Clarifications for the Questions
> Thank you very much for your detailed evaluation, constructive suggestions, and questions! We appreciate the time and effort you have put into reviewing our work. We have **taken your feedback seriously and have made several improvements to the manuscript in the revised version**. We've simplified the wording, enhanced the writing, and incorporated your suggestions and new answers to the questions—such as removing terms like "bespoke." We encourage you to compare the previous and current versions, and we would be grateful for any further suggestions you might have. Below, we address your questions point by point:
>
> > Q1: ... Is there a source for...
> >
>
> **[Q1]** We have added a reference and provided a more detailed discussion of related works in the revised version, specifically in Appendix A.
>
> > Q2: ... Does the Data-Juicer sandbox require specific infrastructure/machines or does it work out-of-the-box? ...
> >
>
> **[Q2]** As a middleware, the sandbox itself does not impose any additional specific hardware dependencies. Instead, it inherits the dependencies of the integrated underlying libraries/frameworks. Besides, to simplify dependencies and avoid redundancy in an "all-in-one" environment, we have introduced and employed a [lazy-loader mechanism](https://github.com/modelscope/data-juicer/blob/main/data_juicer/utils/lazy_loader.py) at the Python package level.
>
> > Q4: Are the 'data processing operators (OPs)' simply data filters?"
> >
>
> **[Q4]** Data-Juicer's operators are not limited to simple data filters; they now encompass a wide range of functionalities, including 100+ Mappers, Filters, Deduplicators, and Selectors. This diversity allows for research on various types of data processing utilities within the Sandbox. For example, Mappers can be employed to examine the effects of data augmentation and editing on downstream model task performance. Given the substantial content already covered in this paper, we have reserved further exploration beyond filters for future work.
>
> > Q5: "Is there a reason why the data is sorted into three equal-sized pools?"
> >
> > Q6: "How is 'size of the data pool' defined?"
> >
> > Q7: "How do you ensure that the models are 'done' training?"
> >
> > Q9: "Is there a fixed computing budget for training a model on each data slice?"
> >
> > Q10: "It may be good to illustrate the training convergence of some of the models on different data slices."
> >
>
> **[Q5-Q7, Q9-Q10]**:
>
> + The number of buckets reflects the trade-off between data intervention intensity (via operator stats) and the reliability of model feedback (the metric changes \delta of post-training downstream tasks). More buckets lead to greater statistical differences between buckets with different ranks (especially the first and last ones), strengthening attribution to data processing effectiveness. However, more buckets also reduce per-bucket data, increasing the risk of inadequate data for models to exhibit reasonable \delta.
> + As a result, we do not aim for models to be "done" or "converged" in this sandbox experiment setting. Instead, we want to observe enlightening changes—positive or negative—in models after targeted data intervention versus random data sampling. In our early experiments, we tested bucket counts of [2,3,4,5] to evaluate whether a model trained on randomly sampled data could reasonably decrease loss after one epoch and show statistically significant changes on downstream tasks. Our findings indicate that three buckets are empirically good for our scenarios. Once determined, all controlled experiments are aligned to one complete epoch and matched to the random pool data size.
>
> > Q8: "How is the 'expansion rate' (L416) defined? If it means duplicating the data slice, please just mention it as such. Duplicating the data slice essentially means that the model has been trained for more epochs. Is it then fair to compare two models when the number of epochs trained is different?"
> >
>
> **[Q8]**: As described in Section 3.3.3 (lines 275~281), "expansion rate" denotes (1) a repeating dataset by duplicating the data slice (red lines in Figure 4) and (2) a non-repeating dataset by consolidating pools from upper to lower pyramid levels to match the iterated dataset size (blue lines in Figures). The comparison is fair because the total data volume passed by both models (red and blue lines) is identical. With each duplication, the blue model sequentially incorporates an equivalent volume from a suboptimal data pool.

---

> ### Author Response · Authors · 2024-11-25
> **Response to Reviewer Wtwc (part 3/3)**
>
> ## [Q11] Considering interaction among data, model and compute
> > Overall, it would be good to consider the interaction between data, models, and compute instead of only the former 2.
> >
>
> Thank you for your suggestion!
> - **Previous interaction**:
>  We would like to point out that our practical examples from two distinct fields, already presented in our paper, *have explored the interaction among the three dimensions*: In Sec. 4.4, we introduce compute scaling by repeatedly training models from 1 to 10 epochs. The red and blue lines represent different dataset constructions, either by duplication or merging distinct but lower-layer data pools, while maintaining the same dataset size.
>
> - **New experiment**:
>  Moreover, within the short rebuttal period, we have swiftly applied the sandbox workflow to a new CLIP model scenario by exploring the interaction from a scaling law perspective. By reusing parts of the datacamp's training and evaluation code, we were able to adapt this quickly and further explore the sandbox's potential in compute scaling, substantiating the suite's adaptability. *More detailed Tables and Figures are provided in the revised Section C.7.* Here we summarize the experiments as follows:
>
>     + `Setting`: We utilize data from the small track of the DataComp competition and adhere to its evaluation metrics, which include 40 distinct evaluation subsets. Due to some broken links, we successfully downloaded 85.2\% of the dataset, resulting in a total of 10.9 million samples as our $\mathcal{D}$. All baseline models were trained on an equivalent volume of data as used in the contrastive experiments, sampled randomly from this dataset. As illustrated in Figure 2, the experiments are also conducted in three phases: first, single-operation processing; second, multi-operation processing; and finally, scaling experiments with *increased data volume across various model sizes and computational scales*.
>     + `Finding Best Recipe`: Table 9 presents the results from single-operation processing. From which we can find that the top three performing operations are _image_text_similarity_filter_, _image_text_matching_filter,_ and _image_nsfw_filter_. In the multi-operation processing phase, we evaluated all possible combinations of the top three operations. The results, illustrated in Figure 11, show that using the _image_text_similarity_filter_ operation alone outperforms other combinations. This observation may be attributed to the filtering process relying on a high-quality CLIP model to train another CLIP model, effectively creating a form of model distillation. Consequently, the _image_text_similarity_filter_ operation consistently dominated the performance outcomes of the other operations.
>     + `Scaling Study`: Finally, we validated whether our findings in the small-scale sandbox experiments could generalize to larger scales from a scaling law perspective. Specifically, we increased the data volume from 880k to 2,683k and examined our conclusions across various models (from ViT-B32 to ViT-B-16) and different computational scales (1/4, 1/2 and 1x relative to the default computation of the DataComp-Small Track). As depicted in Figure 12, the image_text_similarity_filter _operation, identified as the best performer in small-scale experiments, continued to outperform the recipe, which combines _image_text_similarity_filter_ and _image_text_matching_filter_. Furthermore, Figure 12 illustrates notably **consistent improvements when both model and computational scales are increased.** **The linear growth in relative improvement with the exponential increase of computation** aligns with known scaling laws (Cherti et al., 2023), which suggest that expanding model size and data, alongside computational scaling, yields consistent performance gains.
>
> ---
>
> ## Closing Remarks
> Thank you again for your insights and feedback! We respectfully ask you to take another look at the merits of our work by communicating how the platform and insights are integral to one another and essential to the paper's overall contribution. We hope these responses and the new revision can address your concerns and enhance your confidence in the acceptance of this paper. If you have any additional concerns or queries, we warmly invite you to share them with us. Thanks again!

---

> > ### Comment · Reviewer_Wtwc · 2024-11-26
> >
> > Thank you for your comments. I will consider them when adjusting my score.

---

> > > ### Author Response · Authors · 2024-11-27
> > > **Thanks for the reply!**
> > >
> > > Dear Reviewer Wtwc,
> > >
> > > Thank you for your prompt feedback and for considering an adjustment to your score. We appreciate your attention and constructive comments you have provided.
> > >
> > > If there are specific aspects of our response and revision that you feel could be further improved or clarified, please don't hesitate to let us know. We are eager to address any comments and enhance our work. Thank you once again for your time and valuable feedback!
> > >
> > > Best, authors

---

### Official Review · Reviewer_4NsE · 2024-11-02

**Soundness:** 3
**Presentation:** 3
**Contribution:** 2
**Rating:** 6
**Confidence:** 4

**Summary:**

This work introduces an innovative open-source sandbox suite for collaborative development of multimodal data and generative models. IT presents a progressive workflow for data-model development, demonstrating significant performance improvements in image understanding and video generation. Extensive evaluations reveal valuable insights for future multimodal generative models development.

**Strengths:**

- The introduction of an open-source sandbox suite enables more efficient and well-planned data processing and model training processes
- The observations provided are valuable insights that could guide other researchers in their work

**Weaknesses:**

- The paper could benefit from more discussions on how Data-Juicer differs from existing workflows/infrastructure
- I'm also curious to learn whether Data-Juicer is easily adaptable for potential changes/development in the field

**Questions:**

- discussions on how Data-Juicer differs from existing workflows/infrastructure
- Is Data-Juicer easily adaptable for potential changes/development in the field?

---

> ### Author Response · Authors · 2024-11-25
> **Response to Reviewer 4NsE (part 1/3)**
>
> **Dear Reviewer 4NsE,**
>
> We sincerely appreciate your recognition of our work, noting its *"good soundness, presentation, innovative and valuable insights."* Below, we address the two weakness points you raised:
>
> ---
>
> ## [W1 & Q1] More discussion on related works
> > **"The paper could benefit from more discussions on how Data-Juicer differs from existing workflows/infrastructures."**
> >
>
> **Original Placement and Revisions**
>
> + In the submitted version, we placed a brief discussion of related works on the main page, in the penultimate Section. The discussion comprised three aspects:
>     1. _Model-Centric Progress in Multimodal Models._
>     2. _Trends in Data-Centric Development._
>     3. _Open-Source Infrastructure._
> + Due to space constraints, more detailed discussions were in Appendix A (lines 774-827).
> + **Revised Placement:** Per reviewers' suggestions, we moved this discussion earlier, from Section 4 (Page 10) to Section 2 (Page 3), including two new representative works: `DataCamp` and `MetaCLIP`.
>
> **Summarization of the related works**
>
> Here we summarize the discussions and refer to more details and references in the revision file:
>
> 1. **Model-Centric Progress in Multimodal Models:**
>     - **Current State:** Existing works often confine insights to specific datasets or vague data characteristics. This situation leaves a significant gap in comprehending the extent to which models' performance and behavior hinge upon implicit assumptions and inductive biases embedded within the underlying data distributions.
>     - **Our Position:** In contrast, our work demonstrates a feasible and promising path to fill in this gap by explicitly linking data processing effects with the downstream performance of trained models through numerous contrastive sandbox experiments.
> 2. **Data-Centric Development:**
>     - **Current State:** multimodal data processing works mainly involve highly heterogeneous processing workflows, vast quantities, diverse types, and the high cost of training downstream models. This complexity results in predominantly heuristic approaches, such as data filtering and synthesis guided by human intuition.
>     - **Examples:** one well-studied model type is CLIP. `DataComp` introduces a benchmark to filter out high-quality data from 12.8 billion image-text pairs in Common Crawl to train better CLIP models, considering Filter Operators such as CLIP score, image size, and caption length; `MetaClip` aims to reproduce CLIP's data by introducing a raw data pool and finding a balanced subset based on CLIP's concepts.
>     - **Our Position:**
>         1. Unlike these data-centric approaches that isolate the model and training settings, focusing solely on the quality and scale of training datasets, our work emphasizes systematic methodologies for data-model co-development, considering both data and models equally important.
>         2. More specifically, we incorporate performance signals from sandbox reference models on many downstream tasks, conducting importance and correlation analysis to link data pools and these model metrics. Additionally, we explored more model types beyond CLIP, such as LLaVA-like and DiT-based models for image-to-text and text-to-video tasks, and identified better training datasets for these models using our workflow.
> 3. **Open-Source Infrastructure:**
>     - **Current State:** Existing open-source multimodal data infrastructure mainly contributed to certain datasets or dataset-specific preprocessing tools. The standardization and efficient utilization of practical expertise and foundational data processing capabilities remain unaddressed.
>     - **Our Position:**
>         1. Recognizing the critical interplay between datasets and models—where comprehensive, high-quality datasets enhance model performance, and advanced models contribute to the generation of even more refined datasets—our work stands out by ***introducing an innovative intermediary layer.***
>         2. We made substantial effort to integrate cutting-edge model-centric multimodal infrastructure with the Data-Juicer data processing system. This integration fosters a streamlined and insightful co-development environment for both models and data, bridging the current gap and setting a standard for future efforts in the multimodal domain.

---

> > ### Comment · Reviewer_4NsE · 2024-12-02
> >
> > Thank you for your detailed response, I have no further questions.

---

> > > ### Author Response · Authors · 2024-12-03
> > > **Thanks for the reply!**
> > >
> > > Dear Reviewer 4NsE,
> > >
> > > Thank you for your review and for taking the time to consider our detailed response. We appreciate your feedback and are glad to hear that all your questions have been addressed. Should you have any further insights or comments in the future, please feel free to share them. Thank you again for your valuable input!
> > >
> > > Best, authors

---

> ### Author Response · Authors · 2024-11-25
> **Response to Reviewer 4NsE (part 2/3)**
>
> ## [W2 & Q2] Adaptability to other fields
> > **I'm also curious to learn whether Data-Juicer is easily adaptable for potential changes/development in the field.**
> >
>
> Thank you for raising this important question on the adaptability of Data-Juicer Sandbox to potential changes and developments in the field. We appreciate your interest in understanding this aspect, and we would like to address it from two perspectives:
>
> Firstly, from the **code implementation perspective:**
>
> + `Architecture Flexibility`:The design of our sandbox employs a three-layer decoupling approach, consisting of bespoke end-to-end workflows, generic development behaviors, and foundational data-model development capabilities. This architecture allows for rapid and easy adaptation in response to field changes. By clearly defining responsibilities and promoting modularity, our system is easily maintainable and extendable. Each layer interacts only with its immediate upper and lower layers, simplifying development and management while enhancing flexibility and portability.
> + `Dedicated Encapsulation`: Furthermore, we encapsulate changes by unifily exchanging processed dataset, model training weights and evaluation results. We continually integrate representative development libraries and embrace the evolving open-source community. For instance, the Data-Juicer system currently includes over 100 operators covering various modalities, which is double the number of operators available when the initial Sandbox work was completed.
> + `Integration of Open-source Ecosystem`: Additionally, the integrated libraries like modelscope-trainer and LLaVa training codes are also constantly iterating. Our approach thus further accelerates migration and reuse thanks to the community.
>
> Secondly, from the **sandbox operational perspective:**
>
> + `Workflow Generality`: In designing workflows, we strive to maintain the generality of machine learning tasks, encapsulated in four stages: 1) probing data and models (observability implies optimizability), 2) targeted improvements (orchestration of data or training recipes), 3) actual data processing and training, and 4) performance feedback such as on validation sets.
> + `Good Potentials`: The core idea behind this experimental sandbox is to treat data and model parameters as two variables to be optimized. Although this presents a challenging bi-level optimization problem, we posit that the data pool and reference model parameters within the sandbox are small-scale samples of larger application scenarios. Recognizing potential sampling biases and scaling generalization loss, this design shows significant promise in unifying the modeling of different datasets and model architectures to adapt to a broad range of scenarios.
>
> (To be continued.)

---

> ### Author Response · Authors · 2024-11-25
> **Response to Reviewer 4NsE (part 3/3)**
>
> (Continued from previous page, W2 & Q2.)
>
> Lastly, in addition to the practical examples from two distinct fields already presented in our paper, we have, within the short rebuttal period, swiftly applied the sandbox workflow to a **new CLIP model scenario.** By reusing parts of the datacamp's training and evaluation code, we were able to adapt this quickly and further explore the sandbox's potential in compute scaling, substantiating the suite's adaptability. **More detailed Tables and Figures are provided in the revised Section C.7.** Here we summarize the experiments as follows:
>
> + `Setting`: We utilize data from the small track of the DataComp competition~\citep{datacomp} and adhere to its evaluation metrics, which include 40 distinct evaluation subsets. Due to some broken links, we successfully downloaded 85.2\% of the dataset, resulting in a total of 10.9 million samples as our $\mathcal{D}$. All baseline models were trained on an equivalent volume of data as used in the contrastive experiments, sampled randomly from this dataset. As illustrated in Figure 2, the experiments are also conducted in three phases: first, single-operation processing; second, multi-operation processing; and finally, scaling experiments with \textbf{increased data volume across various model sizes and computational scales}.
> + `Finding Best Recipe`: Table 9 presents the results from single-operation processing. From which we can find that the top three performing operations are image_text_similarity_filter, _image_text_matching_filter,_ and _image_nsfw_filter_. In the multi-operation processing phase, we evaluated all possible combinations of the top three operations. The results, illustrated in Figure 11, show that using the _image_text_similarity_filter_ operation alone outperforms other combinations. This observation may be attributed to the filtering process relying on a high-quality CLIP model to train another CLIP model, effectively creating a form of model distillation. Consequently, the _image_text_similarity_filter_ operation consistently dominated the performance outcomes of the other operations.
> + `Scaling Study`: Finally, we validated whether our findings in the small-scale sandbox experiments could generalize to larger scales from a scaling law perspective. Specifically, we increased the data volume from 880k to 2,683k and examined our conclusions across various models (from ViT-B32 to ViT-B-16) and different computational scales (1/4, 1/2 and 1x relative to the default computation of the DataComp-Small Track). As depicted in Figure 12, the image_text_similarity_filter_operation, identified as the best performer in small-scale experiments, continued to outperform the recipe, which combines _image_text_similarity_filter_ and _image_text_matching_filter_. Furthermore, Figure 12 illustrates notably **consistent improvements when both model and computational scales are increased.** **The linear growth in relative improvement with the exponential increase of computation** aligns with known scaling laws (Cherti et al., 2023), which suggest that expanding model size and data, alongside computational scaling, yields consistent performance gains.
>
> ---
>
> ## Closing Remark
> Thank you for the valuable comments and queries you provide. We hope that these responses and clarifications can address your comments, leading you consider an increase in the rating. If you have any additional comments, we warmly invite you to share them with us. Thanks again!

---

### Official Review · Reviewer_jzyi · 2024-11-04

**Soundness:** 2
**Presentation:** 3
**Contribution:** 2
**Rating:** 6
**Confidence:** 4

**Summary:**

The paper introduces a sandbox suite designed for integrated data-model co-development, promoting efficient experimentation. The proposed "Probe-Analyze-Refine" workflow demonstrates notable performance improvements on the VBench leaderboard. Overall, the work presents a practical solution to optimizing multimodal models effectively.

**Strengths:**

- The paper is generally easy to follow, the presentation of the idea is clear.
- Co-developing data + model pipeline is a valid idea and meaningful to explore
- The insights in section 5 are interesting.

**Weaknesses:**

- I found the langauge of the paper often involves exaggeration and flashy vocabulary, and it is too much for an academic paper which sometimes make it hard to read. I suggests authors use more plain and clean language.
- More experiments need to prove the effectiveness of the method, e.g. for the image-to-text tasks, is there any evaluation results demontrating the effectiveness of the method?

**Questions:**

- in section 5.4: "advancing from the 40k data pool used in Section 5 to a significantly more voluminous dataset, approximately 22× larger...", does that mean the performance in tab 2 is achieved with much larger training data than other baselines?
- Is there any ablation done for tab 2, to provide insights on whether the proposed method is data efficient?

---

> ### Author Response · Authors · 2024-11-25
> **Response to Reviewer jzyi (part 1/2)**
>
> **Dear Reviewer jzyi,**
>
> We are immensely grateful for your recognition of the _"clear presentation, validity, meaningfulness, and interesting insights"_ of our work. We sincerely appreciate your constructive suggestions and questions. Below, we address each point in detail:
>
> ---
>
>
> ## [W2] Experiment for Image-to-Text Task
> > Experiment for image-to-text task. ..., for the image-to-text tasks, is there any evaluation results demonstrating the effectiveness of the method?
> >
>
> Regarding your query about the effectiveness of our method for the **image-to-text task**, we have included evaluation results in the revision. **In Table 2 of Section 4.5 (lines 486 ~ 490),** we provide comprehensive experimental results demonstrating the method's efficacy in image-to-text generation. Here, we present a summary of the image-to-text experiments:
>
> + **Setting**: Here we summarize the experiment and more details can be found in the revision. Specifically, we employed Observations 6 and 7 to select a top data pool for pretraining, which we repeated 4 times to create a dataset of 637k samples, approximately half the size of the original pretraining dataset. We then trained the MGM-2B model using this new dataset along with the original fine-tuning dataset.
> + **Results**: As shown in Table 2, despite using only **1/10 of the distinct instances and 1/2 the total instances** of the baseline dataset, our method achieved better performance across various metrics, including MMBench, MMBench-CN, MME-Perception, and MME-Cognition.
> + **Conclusion**: This demonstrates the **data-efficient feature** and effectiveness of our method, highlighting the significant potential for **data-model co-development** in image-to-text models.
>
> ---
>
> ## [Q1 & Q2] Explanation and Ablation for Text-to-Video Experiments
> > "... more voluminous dataset, approximately 22× larger ..., does that mean the performance in tab 2 is achieved with much larger training data than other baselines?"
> >
>
> For the dataset size question, the answer is No:
>
> + **(Clarification on the size)** We would like to clarify the misunderstanding of the _"much larger training data"_ referred to: It is **not compared to other text-to-video baselines** within the Table. Instead, it refers to the **data pool size** in our **previous small-scale sandbox experiments in Section 4**. For clarity, we have **explicitly defined** these data pools and their sizes with notations `$D, D_{random}, P_i, PS$` (see more details in Section 3.3 and Appendix B).
> + **(Size Compared to Baselines)** Despite using **smaller** datasets, our models showed **better performance** than other text-to-video baselines. For example, **VEnhancer** (He et al., 2024a) achieved an average _VBench score of 81.97 with a 350k sample dataset_. In contrast, our models attained a _VBench score of 82.10 with 147k samples and 82.53 with 228k samples_.
>
> > "Is there any ablation done for tab 2, to provide insights on whether the proposed method is data efficient?"
> >
>
> + **(Answer overview)** Yes, we conducted an ablation study to analyze data efficiency, detailed in Appendix Section C.5 (lines 1289 ~ 1398 in the revision). We focused on modifications to T2V-Turbo, which included enhancing the data pool, initializing parameters for LoRA, self-distillation, adding real-data loss, and self-evolution. Throughout these progressive modifications, we observed continued improvements in **quality** and **semantic scores** of both training data and generated videos, ultimately establishing a **new state-of-the-art performance** based on the co-development process.
> + **(Summarized exp settings and results)** We briefly summarize these experiments and refer to more details in the Revision. Initially, by replacing the WebVid data with our proposed data pool of 147k instances, the total score improved from 81.01% to 81.84%, indicating better data efficiency. Further ablation studies showed that initializing LoRA with the T2V-Turbo model did not enhance performance, possibly due to the effectiveness of T2V-Turbo as compared to the teacher model. Our self-distillation method raised semantic scores but led to unstable video quality with temporal flickering. We further integrated a real-data loss to maintain video quality and encourage the trained model aware of the refined dataset. Subsequently, through self-evolution, we expanded the data pool by an additional 228k instances to further evolve the model.

---

> ### Author Response · Authors · 2024-11-25
> **Response to Reviewer jzyi (part 2/2)**
>
> ---
> ## [W1] Writing Suggestions Regarding Vocabulary Style
> Many thanks for your suggestion about the vocabulary style. We have tried our best to simplify the wording and improve the writing in the revision, such as replacing "tapestry" with "diversity". You can compare the previous version for any inappropriate style or wording. We would be grateful if you could point them out to us.
>
> ---
>
> ## Closing Remark
> Thanks for the valuable comments and queries you provided. We hope that you can consider an increase in the rating if these responses and clarifications address your comments. If you have any additional comments, we warmly invite you to share them with us. Thank you again!

---

> ### Author Response · Authors · 2024-12-03
> **Kind Reminder to Follow-up on Experimental Results and Revisions**
>
> Dear Reviewer jzyi,
>
> As the discussion deadline approaches, we notice that you have not yet replied to our response. We have explained the experimental results of the image-to-text task and the ablation studies of the text-to-video task. Additionally, we have improved our writing in the revision according to your suggestions. We sincerely hope that our response has addressed your questions.
>
> May we kindly ask if you have any further questions regarding this matter? We would greatly appreciate it if you could provide us with your feedback at your convenience. Thank you very much for taking the time to review and consider our comments.
>
> Best, authors.

---

### Author Response · Authors · 2024-11-30
**Gentle Reminder Regarding the Author-Reviewer Discussion Deadline**

Dear Reviewers,

We sincerely appreciate your time and effort in reviewing our paper, as well as the recognition and valuable feedback you have provided!
As the discussion phase will conclude in a few days, we have received only one pending response.

We have carefully considered all your comments and made every effort to **directly address each raised weakness and question**. For your convenience, we summarize the responses and revisions below, which may assist in providing a quick tour and facilitate a comprehensive assessment:

`More Experimental Evidence`

1. *(Reviewers AxLt, Wtwc, 4NsE)* We quickly applied the sandbox workflow to a new **CLIP model scenario** and further explored its potential in **compute scaling**. Additionally, we summarized how the previous image-to-text and text-to-video experiments explore interactions between *data, model, and compute*. This substantiates the suite's adaptability and effectiveness.

2. *(Reviewer jzyi)* We included **larger-scale** experiments for the **image-to-text task** in the revised paper. *Table 2* in *Section 4.5* demonstrates the **data efficiency** and effectiveness of the proposed method.

3. *(Reviewer jzyi)* We provided a detailed **ablation study** of the text-to-video model that ranks at the top of the VBench board. This is covered in *Section 3.3* and *Appendix B*, quantitatively showing why our method performs well on the **text-to-video task**, and highlighting its data efficiency.

`Clarifications of Reviewer Queries and Misunderstandings`

1. *(Reviewers AxLt, 4NsE)* We have included a comparative discussion and incorporated the reviewer's recommended works in the **related work section**, with more details added in our revision.

2. *(Reviewers Wtwc, AxLt)* We have provided a structured discussion about the paper's focus, balancing "Platform vs. Insight", and have **improved the organization** of the paper according to your suggestions.

3. *(Reviewers Wtwc, jzyi)* Efforts have been made to **simplify the language and improve clarity** in the revised paper, following your suggestions on writing style and questions regarding some implementation details.

We believe that these enhancements have strengthened our paper, thanks to your insightful comments. If you have further questions or require additional clarification, please feel free to reach out.

*We kindly request you to review these responses and re-evaluate the merits of our work*. Your feedback is highly anticipated and greatly valued. Thank you once again!

Best, authors

---

### Meta-Review · Area_Chair_UUQX · 2024-12-23

**Metareview:**

This paper presents Data-Juicer Sandbox, a comprehensive suite for integrated data-model co-development in multimodal generative models. The key claims are that it enables systematic data-model optimization through a "Probe-Analyze-Refine" workflow, demonstrated through improved performance on tasks like image-to-text generation and text-to-video generation. While the paper shows strong empirical results, including topping the VBench leaderboard, several critical weaknesses make it unsuitable for acceptance at ICLR at this time. The paper lacks clear systematic validation of its claims - there is insufficient analysis of compute costs and scaling behavior across different model sizes and architectures. The experimental methodology has gaps in controlling for compute budgets when comparing different approaches. The organization buries important technical details in supplementary material rather than presenting them clearly in the main paper. Though the authors added CLIP experiments during rebuttal and provided more compute analysis, fundamental questions remain about whether the observed improvements come from better data processing or simply from using more compute. Based on these concerns I vote to reject.

**Additional Comments On Reviewer Discussion:**

The reviewers raised several significant concerns that led to productive discussion. Reviewer AxLt highlighted issues with compute analysis and scaling behavior - while the authors added CLIP experiments and compute estimates, questions remain about proper computation of training FLOPs and derivation of scaling laws. Reviewer Wtwc questioned the paper's focus and organization - the authors clarified the dual platform/insights contribution but didn't fully address concerns about buried technical details. Reviewer jzyi requested more experimental results and ablations - these were provided but raised new questions about fair comparisons. Reviewer 4NsE asked about adaptability - the authors demonstrated this through new CLIP experiments but didn't fully address generalization concerns. Though the authors made a good faith effort to address concerns through additional experiments and analysis, fundamental questions about rigorous validation remain unresolved. The key issues around compute-controlled comparisons, scaling analysis, and clear presentation of technical details were not adequately addressed despite multiple rounds of discussion.

---

### Decision · Program_Chairs · 2025-01-22

Reject